# STAR: Synthesis of Tailored Architectures

Armin Thomas[1], Rom Parnichkun[1,2], Alexander Amini[1], Stefano Massaroli[1,3], and Michael Poli[1,4]

[1]Liquid AI [2]The University of Tokyo [3]RIKEN [4]Stanford University
*{armin,rom,alexander,stefano,michael}@liquid.ai*

## Abstract

Iterative improvement of model architectures is fundamental to deep learning: Transformers first enabled scaling, and recent advances in model hybridization have pushed the quality-efficiency frontier. However, optimizing architectures remains challenging and expensive. Current automated or manual approaches fall short, largely due to limited progress in the design of *search spaces* and due to the simplicity of resulting patterns and heuristics. In this work, we propose a new approach for the *synthesis of tailored architectures* (STAR). Our approach combines a novel search space based on the theory of *linear input-varying systems*, supporting a hierarchical numerical encoding into architecture *genomes*. STAR genomes are automatically refined and recombined with gradient-free, evolutionary algorithms to optimize for multiple model quality and efficiency metrics. Using STAR, we optimize large populations of new architectures, leveraging diverse computational units and interconnection patterns, improving over highly-optimized Transformers and striped hybrid models on the frontier of quality, parameter size, and inference cache for autoregressive language modeling.

## 1 Introduction

Most domains of applications for AI have seen a gradual convergence towards similar model architecture designs, based on stacking multi-head attention and gated linear units (Transformers) (Vaswani et al., 2017; Shazeer, 2020; Brown, 2020) or combinations of other basic computational units grounded in signal processing, such as recurrences or convolutions (Martin & Cundy, 2017; Romero et al., 2021; Gu et al., 2022; Smith et al., 2023; Peng et al., 2023; Poli et al., 2023; Massaroli et al., 2023; Yang et al., 2024b).

Broadly, there are two prominent paths to improve model architectures: automated and manual. Automated design, leveraging optimization (e.g. evolutionary methods) within a predefined search space, has seen success in highly-targeted domains, such as the refinement of convolutional neural networks for resource-constrained applications (Pham et al., 2018; Liu et al., 2018; Howard et al., 2019; Li et al., 2021; Tan & Le, 2021). Automated methods have also been utilized to identify candidate improvements to standard computational primitives (So et al., 2021), e.g., depthwise convolutions in projections. Nevertheless, to date, automated methods have fallen short of providing a unified framework that provides significant improvements in **quality** and **efficiency** across domains and objectives over models using standard generalizable recipes. The homogeneity of architectures applied at scale during the Transformer era highlights this shortcoming.

The main challenge for automated methods lies in curating a *search space* for computational units and architectures that is both (a) *well-conditioned* i.e., populations of model candidates can be trained effectively, without numerical instability or unpredictable degradation in performance, and (b) *comprehensive* i.e., the design space includes candidates with significantly different properties from existing variants, expanding the range of potential improvements that can be identified.

Despite a wealth of automated approaches for the search and refinement of computational units and composition strategies (White et al., 2023), the current generation has been obtained mostly through an iterative manual process, guided by intuition and tuning on representative smaller-scale tasks via e.g., synthetics and scaling laws (Hoffmann et al., 2022; Arora et al., 2023; Bi et al., 2024). Manual design has led to a variety of results, most notably in the introduction or improvement of computational units (Poli et al., 2023; Massaroli et al., 2023; Yang et al., 2024a; Arora et al., 2024),

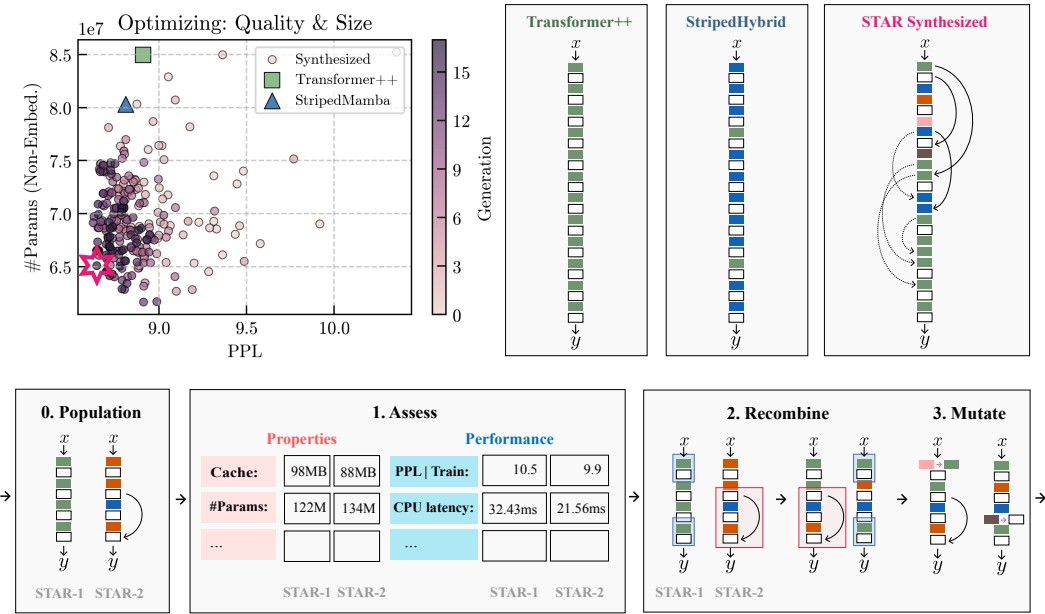

Figure 1.1: **[Top Left]:** Population of architectures undergoing iterative `STAR` evolution to minimize number of parameters and maximize quality. **[Top Right:]** Baseline Transformer++, hybrid model, and representative architecture found via `STAR`. **[Bottom]:** `STAR` evolution optimizes architectures using principles of evolutionary optimization, including assessment, recombination, and mutation.

modifications targeting smaller inference cache (Shazeer, 2019; Brandon et al., 2024), and the discovery of simple interconnection strategies e.g., *striped* hybridization (Brown, 2020; Fu et al., 2022; Poli et al., 2024; Lieber et al., 2024), weight sharing (Liu et al., 2024), and others (Liu et al., 2022; Brandon et al., 2024). Yet, manual design is limited to finding relatively basic design patterns, compared to the total diversity of possible patterns, and requires a significant investment in resources, expertise and time.

Given the wide range of possible applications of current AI systems, enabling systematic and automatic optimization of model architectures from the multitude of existing computational units is key to meeting the various demands these applications pose, in terms of efficiency (e.g., model size, inference cache size, memory footprint) and quality (e.g., perplexity, downstream benchmarks), and a prerequisite on the path to further, consistent improvements on the quality-efficiency Pareto frontier.

In this work, we seek to address limitations of existing automated architecture optimization methods by introducing an approach for the *synthesis of tailored architectures* (`STAR`). `STAR` is based on the combination of a novel hierarchical search space of computational units and their composition, as well as a numerical encoding compatible with evolutionary methods.

**Hierarchical search spaces**   The design space for `STAR` is grounded in the theory of *linear input-varying systems* (LIVs), providing a novel framework to design building blocks in architectures. LIVs generalize most computational units used in deep learning, such as attention variants, linear[1] recurrences, convolutions, and other structured operators. Notably, our framework allows us to characterize model architectures at three hierarchical levels of resolution: (a) *featurization*, defining how the linear computation within the LIV is modulated by the input context; (b) *operator structure*, defining the token and channel mixing structure of the LIV; and (c) *backbone*, defining the composition structure between LIVs. In contrast to previous search spaces (Pham et al., 2018; Liu et al., 2018; Howard et al., 2019; Li et al., 2021; Roberts et al., 2024), we show how the LIV search space is both comprehensive and well-conditioned, as most sampled candidates train without instabilities.

Leveraging the modularity of LIVs, we taxonomize the space and devise a hierarchical numerical representation of a model backbone, which we refer to as the `STAR` genome. Due to its struc-

---

[1]Here, by *linear* we refer to the linearity of the state transition.

ture, STAR genomes can be optimized at different levels of the hierarchy. We show how backbone genomes – defining ordering and interconnection between LIVs – can be automatically refined with evolutionary algorithms relying on few key principles, such as evaluation, recombination, and mutation.

**Exploring the efficiency-quality frontier**   We evaluate STAR on autoregressive language modeling, a domain historically dominated by Transformers and architecture improvements found via manual search. In particular, we optimize architectures for various combinations of metrics simultaneously: *quality* (perplexity during pretraining), *quality and size*, and *quality and inference cache size* (KV cache and fixed state cache). When optimizing for quality and size, 7/8 evaluated STAR-evolved architectures improve over Transformer++ and striped hybrids of recurrences and attention across downstream evaluation benchmarks (Gao et al., 2024), with a reduction of up to 13% in parameter counts. Similarly, optimizing for quality and cache size, 7/8 evaluated STAR-evolved architectures achieve up to 37% smaller cache sizes than striped hybrids, and 90% smaller than Transformers, while performing at least as well in quality. We also show that 125M-parameter architectures optimized for quality and cache by STAR can scale to 1B parameters and perform on par with parameter-matched Transformer++ and striped hybrid architectures, while maintaining the same advantages in cache size reductions. When optimizing solely for quality, all evaluated STAR-evolved architectures outperform standard hybrids on downstream benchmarks, achieving improvements twice as large as those of hybrids over Transformers. Finally, we showcase how STAR can be used to identify recurring *architecture motifs* emerging during evolution, driving the observed performance gains.

## 2   FOUNDATIONS OF THE SEARCH SPACE

We detail how the framework behind our search space is grounded in the theory of linear systems.

**Linear input-varying systems**   The class of data structures under consideration are sequences of vectors $\{x_0, x_1, \ldots, x_\ell\}$ where each element $x_i$ is referred to as a *token*. Each token $x_i$ is a real-valued vector in $\mathbb{R}^d$, represented as $x_i = (x_i^0, x_i^1, \ldots, x_i^{d-1})$. The individual components $x_i^\alpha$ of each token are called *channels*.

The attention operator (Bahdanau et al., 2014; Vaswani et al., 2017), provides a valuable starting point to define a search space for model architectures, as it defines a prototype of what we call *linear input-varying* (LIV) operators. Attention, in its common formulation, can be expressed as a linear operator applied to the input, with the operator's action determined by the input itself:

$$y_i^\alpha = \underbrace{\sum_{\beta \in [d]} \sum_{j \in [\ell]} \sigma(q_i^\top k_j) V^{\alpha\beta}}_{\texttt{attention operator}} x_j^\beta, \quad (q_i, k_i) = (\varphi(x_i), \psi(x_i))$$

where $\sigma : \mathbb{R} \to \mathbb{R}$ is a nonlinear function and $V \in \mathbb{R}^{d \times d}$. The intermediate quantities $q, k$, obtained through functions $\varphi, \psi : \mathbb{R}^d \to \mathbb{R}^h$ of the input tokens and used to construct the linear operator $T$, are referred to as *feature groups*.

We extend this idea to include a broader family of LIVs, expressed in their most general form as:

$$y_i^\alpha = \sum_{j \in [\ell]} \sum_{\beta \in [d]} T_{ij}^{\alpha\beta}(x) x_j^\beta.$$

The LIV framework decouples the (potentially) nonlinear and linear computation required to materialize the operator $T(x)$ and apply it to obtain the outputs, $y = Tx$. LIVs include and generalize a diverse array of computational units commonly used in model architectures, whose class is defined by the structure of the operator: attention, convolutions, linear recurrences, and various forms of other structured layers:

$$
\begin{array}{llr}
T_{ij} = \sigma(C_i B_j) & \textit{dense} & \text{attention [4; 45]} \\
T_{ij} = C_i B_j & \textit{low-rank} & \text{linear attention  [21]} \\
T_{ij} = C_i A_{i-1} \cdots A_{j+1} B_j & \textit{semi-separable} & \text{linear recurrence [28; 41; 17; 49]} \\
T_{ij} = C_i K_{i-j} B_j & \textit{scaled Toeplitz} & \text{gated convolution  [12; 33]} \\
T_{ij} = \begin{cases} \sigma(C) & i = j \\ 0 & \text{otherwise} \end{cases} & \textit{diagonal} & \text{memoryless system  [39]}
\end{array}
$$

where the *structure*[2] of the operator $T$ induces a decomposition into feature groups, analogously to the attention example. To highlight the parallels between different LIVs and attention, we have adopted a shared notation for the feature groups. Differentiating between LIV systems are two key factors, operator *structure* and *featurization*.

**Featurization**    refers to the process with which feature groups are obtained, either via direct parametrization, reparametrization[3], or via parametric transformations of inputs as is the case in attention i.e., linear projections.

**Structure**    We taxonomize the linear operators of LIVs by decoupling the analysis of token-mixing and channel-mixing structures. To define structure, we look at two different *slices* of the operator:

- $T^{\alpha\beta} \in \mathbb{R}^{\ell \times \ell}$ highlights the token-mixing structure for each tuple of input and output channels, i.e. the linear contribution of the $\beta$th input channel $x^\beta \in \mathbb{R}^\ell$ to the $\alpha$th output channel $y^\alpha \in \mathbb{R}^\ell$. Loosely speaking, the choice of token-mixing structure determines the class of matrix multiplication algorithms that can be utilized to apply the operator to the input (e.g. *Fast Fourier Transform* based convolution if Toeplitz, or *parallel prefix scan* if *semi-separable* (Dewilde & Van der Veen, 1998)).

- $T_{ij} \in \mathbb{R}^{d \times d}$ conversely reveals the channel-mixing structure of $T$, i.e. the (linear) contribution of the $j$th input token $x_j$ of the input sequence to the $i$th output token $y_i$. In practice, the most common choice of channel-mixing structure is by far the diagonal one, as used in attention and most variants of linear recurrences[4]. Diagonal $T_{ij}$ blocks allow maximum parallelization of the LIV operators as the linear computation reduces to $d$ independent matrix multiplications.

Note that we use the same logic to define the structure present in the featurizer itself. The structure of the featurizer and operator need not be the same.

**Composition**    An architecture backbone can be decomposed into a set of LIVs with different composition rules. Beyond sequential stacking of LIVs, as is common in standard deep architectures, we introduce other composition rules realized via the featurization: *featurizer sharing*, where the same featurizer weights are shared between different LIVs of a backbone, and *feature group sharing*, where different LIVs share the same feature groups.

Let $T, S$ denote the operators at two different depths $m, n$ ($m < n$) of the composition, respectively. If, for example, both LIV operators are chosen with similar low-rank (linear attention) structure $T_{ij} = C_i B_j$, $S_{ij} = E_i F_j$ and the dimensions of the feature groups are compatible, i.e. $C_i, E_i \in \mathbb{R}^{d \times h}$ and $B_i, F_i \in \mathbb{R}^{h \times d}$, we can apply both featurizer and feature group sharing techniques:

- If the parametric featurizer of $C_i$ and $E_i$ has the same form $C_i = \varphi(x_i; \cdot)$, $E_i = \varphi(x_i; \cdot)$, we can share the same set of parameters $\theta$ between them:

$$T_{ij} = \varphi(x_i^{(m)}; \theta)B_j \quad \underset{\text{featurizer sharing}}{\rightsquigarrow} \quad S_{ij} = \varphi(x^{(n)}; \theta)F_j$$

where $x^{(m)}, x^{(n)}$ denote the input to the $m$th and $n$th LIV system, respectively.

- Similarly, we can simply re-use one of the feature groups of the $m$th LIV system in the $n$th one, e.g. $B_j$:

$$T_{ij} = C_i B_j \quad \underset{\text{feature group sharing}}{\rightsquigarrow} \quad S_{ij} = E_i B_j$$

A prominent example of feature group sharing is the sharing of key-value caches between attention operators (Brandon et al., 2024). Beyond featurizer interconnections, we explore other strategies of operator composition in Appendix A.7.

---

[2]Note that in this list we refer to the sequence-mixing structure i.e., the structure of *slices* $T^{\alpha\beta}$ of the operator $T$. We define the structure via entries $T_{ij}^{\alpha\beta}$ for convenience.

[3]Sometimes referred to as implicit parametrization (Mescheder et al., 2019; Romero et al., 2021; Sitzmann et al., 2020).

[4]This is brought to an extreme by *multi-headed* architectures that only present number-of-heads distinct values on the diagonal.

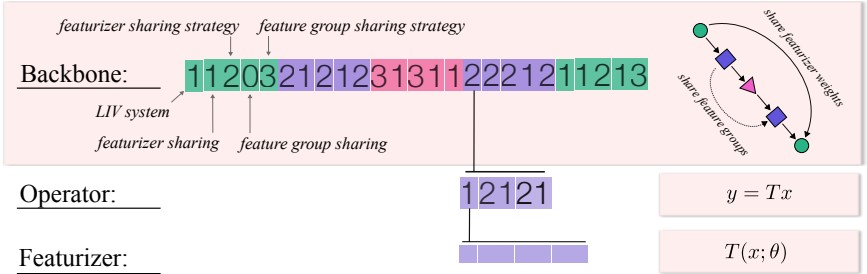

Figure 3.1: Hierarchical structure of the STAR genome. Each sequence at lower levels is summarized into a single value at higher levels, enabling its treatment as a discrete variable. We leverage this property extensively when optimizing backbones directly.

## 3 DESCRIBING OPERATORS AND BACKBONES WITH STAR GENOMES

The new design space of LIVs and their compositions serves as the foundation for the automated *synthesis of tailored architectures* (STAR). In the following, we will describe how we map the three hierarchical levels of the LIV description–featurization, structure, and composition–into a numerical representation suitable for optimization: the *STAR genome*. Each level of the hierarchy can be summarized into a single integer, yielding a numerical representation that can be optimized at different levels of granularity (see Fig. 3.1). In this work, we focus on backbone optimization but are reporting the full description of the genome for completeness. See Appendix A.6 for a full description of all genome values considered in this work.

### 3.1 BACKBONE GENOME

We begin by describing the highest abstraction level of the STAR genome, the *backbone genome*, which defines the composition of LIVs in the backbone. We recall that under the LIV framework, LIVs can be connected with featurizer and feature group sharing, as described in Section 2. Specifically, the backbone genome represents a set of integer-valued sequences of length five, one for each LIV of the backbone. Each of these 5-number segments defines the following properties of the LIV:

1. **LIV class**: integer summary of lower levels of the STAR genome, i.e., operator and featurizer genomes (see Section 3.2).

2. **Featurizer sharing**: determines the weight-sharing structure between featurizers of LIVs at different depth in the backbone. LIVs with the same value at this position share featurizer weights, as defined by the *featurization sharing strategy*.

3. **Featurization sharing strategy**: defines how featurizer sharing is implemented for the LIV class. Featurizer weights can be shared partially, for example, only those responsible for computing $B(x)$, in contrast to also sharing weights that compute $C(x)$. We explore all combinations of sharing strategies based on the number of feature groups of the LIV class.

4. **Feature group sharing**: LIVs with the same index share feature groups directly, instead of featurizer weights, for example, by using the exact same $B(x)$ and $C(x)$.

5. **Feature group sharing strategy**: describes which feature groups, of all available feature groups of the LIV class, are shared.

These 5-number segments are then repeated in the order at which the encoded operators occur in the backbone (Figure 3.1). Note that outside of the compositions defined by the backbone genome, LIVs are sequentially stacked using pre-norm residuals i.e., $y = T(\texttt{norm}(x))\texttt{norm}(x) + x$.

**Example:** `21211-31112-21221-32112` is the backbone of a genome with 4 LIVs. The first and third LIV belong to the same class "2", and are part of the same featurizer sharing group "1", thus sharing featurizer weights according to the indexed featurizer sharing strategy "2" (e.g., only sharing the weights responsible for the first feature group). The second and fourth LIV belong to class "3", they do not share any featurizer weights (they have different featurizer sharing indices, "1" and "2"), and instead share feature groups directly, with strategy "2" (sharing all groups).

Figure 4.1: Fundamental operations of STAR evolution (akin to other evolutionary optimization algorithms).

## 3.2 OPERATOR AND FEATURIZER GENOMES

In the backbone genome, LIVs are summarized into a single number, indicating the specific featurizer and structure of the LIV. Unrolling this encoding reveals an additional level, the *operator genome*, which identifies a specific LIV in 5 numbers: (1) **featurization**, indicating the specific featurizer class; (2) linear **token-mixing structure** of the LIV ($T^{\alpha\beta}$); (3) possible **structured sparsity masks** (e.g., banded) for token-mixing; (4) any **nonlinearity** applied to the token-mixing structure; (5) the LIV's **channel-mixing structure** ($T_{ij}$).

The featurizer class can be similarly unrolled. In STAR, each *featurizer genome* is a sequence defining for each of the feature groups: token and channel mixing structure (akin to the operator genome); expansion factor, defining the ratio of the feature group channel dimension over the input channel dimension, encoded as one number; repeat factor, defining how many times the feature groups are replicated across the channel dimension, encoded as one number. For a detailed description of all genome entries, and the respective values considered in this work, see Appendix A.6.

## 4 SYNTHESIZING ARCHITECTURES BY EVOLVING GENOMES

We have devised the STAR genome as a hierarchical numerical representation that encodes a specific LIV backbone, suitable for gradient-free optimization. In the following, we will outline how a STAR genome can be optimized via evolutionary methods (Beyer & Schwefel, 2002); a process subsequently referred to as *STAR evolution*. To allow for the application of evolutionary optimization methods to the STAR genome, we adapted methods commonly used to iteratively evolve an initial population of genomes.

### 4.1 KEY STEPS OF STAR EVOLUTION

**Assessment**    STAR evolution begins by evaluating the quality of each genome in the initial population. This involves realizing the model encoded in each genome and scoring it against the objective functions of interest, either by training and assessing its performance or through a static analysis for efficiency objectives, such as the total number of trainable parameters in the model. Notably, STAR evolution can incorporate multiple and diverse objectives.

**Pairing**    After assessing the quality of each genome, STAR evolution selects parent genomes for generating the next generation of offspring through tournament selection. Parents are chosen by randomly selecting a set of $k$ genomes from the population and then picking the one with the highest quality—typically based on criteria such as predictive accuracy or the lowest parameter count.[5].

**Recombination**    Next, STAR evolution generates new candidate solutions by applying $k$-point crossover to the selected parent genomes. Here, genetic material from two parents is exchanged between $k$ randomly chosen points, resulting in offspring that inherit traits from both parents. All random sampling in STAR is performed with a uniform probability across all valid options.

**Mutation**    Finally, STAR evolution introduces random mutations to the offspring. These mutations are essential for maintaining diversity in the population and promoting exploration of the search space. In STAR evolution, random mutations are implemented as alterations to the numbers in a genome, where values are randomly replaced by others from a predefined set of possible choices. As

---

[5]As we will explore later, STAR evolution also takes into account solution diversity in the selection process to maintain variety within the population (see Appendix A.4)

discussed earlier (Sec. 3), these choices vary depending on the genome position. To ensure that all genomes encode models capable of being trained stably and showing smooth quality improvements, STAR enforces several constraints on these random mutations, as outlined below (Sec. 4.2).

## 4.2 GUIDING EVOLUTION WITH HIERARCHICAL MUTATIONS

To ensure compatibility with evolutionary optimization, the STAR genome must remain robust to random edits such as recombination, mutation, or initialization. The backbone genome is composed of 5-number segments. When mutating the first entry (LIV class), mutation is restricted to valid LIV classes. For the second and fourth entries (defining featurizer and feature group sharing), only LIVs within the same class can connect. The third and fifth entries are mutated by randomly sampling valid sharing strategies for the corresponding LIV class. If mutations or recombinations result in invalid configurations, such as incompatible sharing strategies, these are detected and repaired by either removing the invalid connection for entries 2 and 4 of the operator genome, or re-sampling the respective genome value from the set of valid choices for entries 1, 3, and 5.

**Controlled improvements** Combining the LIV search space, genome encoding, and guidelines for mutation and recombination, leads to stable training runs for most candidates obtained during the course of STAR evolution, as shown in Figure 4.2.

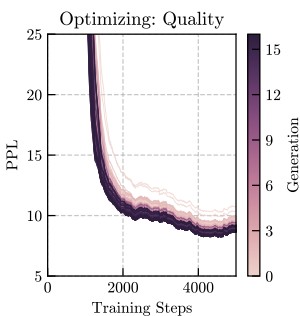

Figure 4.2: Training perplexity for all runs during STAR evolution of a population.

## 5 EXPERIMENTS

**Experimental setup** The goal of our experiments is to test whether STAR is suitable for synthesizing architectures tailored to diverse objectives, such as predictive quality and efficiency. If not noted otherwise, STAR evolutions presented are performed at 125M-parameter model scale, where backbones contain 24 LIVs at a width of 768 dimensions, with populations of 16 genomes that are evolved for 18 generations. During each STAR evolution, we keep the depth and width of the backbone fixed. Experiments are performed in autoregressive language modeling on 4096 token sequences from the RedPajama dataset (Weber et al., 2024).

**Training details** During STAR evolution, models are trained from scratch for 1.3B tokens using AdamW (Loshchilov et al., 2017) with a peak learning rate of 0.0008, a batch size of 0.25M tokens, and a cosine learning rate schedule with a 130M-token linear warmup. The resulting synthesized backbones are evaluated by training them from scratch for 5B tokens under the same setup but with an extended warmup of 400M tokens. Additionally, we train select 1B-parameter models (48 LIVs at a width of 2048) for 40B tokens, increasing the batch size to 0.75M tokens and the warmup to 2.6B tokens. See Appendix A.2 for details.

**Evaluation** We use a two-stage evaluation process. During STAR evolution, performance metrics are computed on a 500M-token evaluation set from RedPajama (Weber et al., 2024)[6]. Post-evolution, we select the 8 models with the lowest perplexity among those with lower parameter counts (for quality and quality-size optimizations) or smaller cache size (quality-cache optimization) than baseline models. These models are trained further and evaluated on downstream tasks: HellaSwag (Zellers et al., 2019), ARC-e (Clark et al., 2018), Winogrande (Sakaguchi et al., 2019), PiQA (Bisk et al., 2020), and SciQ (Welbl et al., 2017); and ARC-c (Clark et al., 2018) for 1B-parameter models.

**Option pool** To improve initialization during STAR evolution, we incorporate genomes of common backbone types (see Sec. 4). Unless stated otherwise, initial populations include simple hybrid backbones without special interconnections, combining memoryless LIVs (e.g., SwiGLU (Shazeer, 2020)) with baseline LIVs such as convolutions, recurrences, or attention. Other backbones are randomly initialized, with random LIV class choices and compositions. As the focus is on backbone optimization, the pool of LIV classes is limited to a subset of systems encodable through the operator and featurizer genomes, including token-mixing structures from Section 2. This includes common dense channel-mixing featurizers (linear projections), more advanced Toeplitz token-mixing featurizers, and "differential" variants of all LIV classes except memoryless ones, which use two identical, parallel LIVs and output their difference. See Appendix A.6 for all included LIV classes.

---

[6]We used two different randomly-drawn evaluation datasets for our ablation and main experiments.

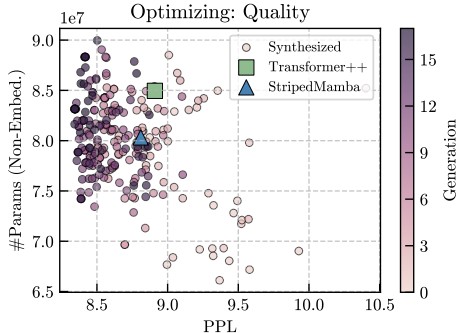

Figure 5.1: **[Left]:** Evolutionary algorithms: Final populations evolved with the Firefly Algorithm (FA), Genetic Algorithm (GA), and Non-dominated Sorting Genetic Algorithm II (NSGA-2). **[Right]:** Backbone synthesis scales (left to right): synthesized at reduced depth ("motif," 8 LIVs at 768), reduced width (24 LIVs at 256), or full depth and width (24 LIVs at 768). Models are scaled to the same LIV count and width via stacking or width extension.

## 5.1 IDENTIFYING A SYNTHESIS PROTOCOL

**Evolutionary algorithm** We compare three gradient-free evolutionary algorithms—Firefly Algorithm (FA) (Yang, 2009), Genetic Algorithm (GA) (Bremermann et al., 1966), and NSGA-2 (Deb et al., 2002)—for optimizing STAR genomes toward quality and parameter count. Each algorithm evolves a population of 16 genomes (8 LIVs, 768 dimensions) over 8 iterations (details in Appendix A.4)[7]. Results show GA and NSGA-2 outperform FA, achieving lower parameter counts while maintaining comparable predictive quality. GA slightly surpasses NSGA-2 in performance but produces larger models, whereas NSGA-2 achieves greater solution diversity (Fig. 5.1). We thus use NSGA-2 in subsequent experiments. Hyperparameter tuning indicates optimal performance with a population size of 16, a 10% mutation probability, and 2 crossover points (Appendix A.4)[8].

**Synthesis scale** Automated architecture optimization in language modeling faces the challenge of high compute costs for training and evaluating large-scale models. We explore two paths to reducing this cost: (a) optimizing smaller backbone *motifs* (groups of fewer LIVs in deeper models) and stacking them to build deeper models; and (b) optimizing full-depth backbones at reduced widths. For both approaches, we evolve 16 genomes over 12 iterations, optimizing for parameter count and quality, and compare the resulting models to those synthesized at full width and depth under identical settings. From each evolution, we select 8 backbones smaller in parameter count than Transformer++ and StripedMamba baselines (Appendix A.3) and with the lowest evaluation perplexity. Selected backbones are scaled to the same LIV count and width (via motif stacking or width extension) and trained for 5B tokens before downstream evaluation.

> **Finding 1:** Synthesizing backbones at full width and depth yields consistent improvements, while reduced-width synthesis achieves similar results with fewer successful candidates. Motif synthesis underperforms both approaches (Fig. 5.1).

## 5.2 SYNTHESIZING HIGH-QUALITY LANGUAGE MODELS

We apply our identified protocol to synthesize high-quality backbones for language modeling by evolving a population using perplexity as the only objective. When optimizing for quality, STAR evolution achieves a reduction of the average quality of an initial population by 1.0 PPL point without changes to model depth and width (Figs. 5.2, B.34).

> **Finding 2:** STAR backbones–optimized for quality–outperform parameter-matched Transformer++ and StripedMamba backbones in RedPajama eval. PPL as well as on Hellaswag, ARC-Easy, Winogrande, PiQA, and SciQ. Improvements of STAR backbones over standard hybrids on benchmark averages is 2 times larger than the improvement of hybrids over Transformers (Tables 5.1 and A.4).

Figure 5.2: Genome scores during STAR evolution, when optimizing for quality.

---

[7]FA and GA optimize a single objective, using the sum of normalized loss $L$ and parameter count $P$: $U(L) + U(P)$, where $U(x) = \frac{x - \min(X)}{\max(X) - \min(X)}$ for $x \in X$.

[8]The 2 best-performing genomes per population are carried over to prevent performance regression.

| Backbone / Optimized for | Size | Cache (bytes \| 4K) | RedPj. ppl ↓ | Hella. acc. norm. ↑ | ARC-e acc. ↑ | Wino. acc. ↑ | PiQA acc. ↑ | SciQ acc. ↑ | Avg. ↑ |
|---|---|---|---|---|---|---|---|---|---|
| Transformer++ | 85M | 150MB | 7.3 | 28.9 | 38.8 | 51.2 | 61.2 | 64.1 | 48.8 |
| StripedMamba | 80M | 25MB | 7.2 | 28.6 | 39.3 | 51.1 | 60.9 | 67.4 | 49.5 |
| STAR-1 / Quality | 79M | 100MB | 7.0 | 29.8 | 39.3 | 51.2 | 62.2 | 72.5 | **51.0** |
| STAR-2 / Quality | 80M | 82MB | 7.1 | 29.2 | 40.5 | 51.1 | 61.6 | 72.4 | **51.0** |
| STAR-3 / Quality | 78M | 120MB | 7.1 | 29.7 | 40.0 | 50.9 | 62.0 | 71.2 | **51.0** |
| STAR-4 / Quality | 79M | 94MB | 7.1 | 29.3 | 39.7 | 51.0 | 61.5 | 72.6 | 50.8 |
| STAR-1 / Q.+Size | 74M | 63MB | 7.2 | 28.9 | 39.3 | 51.0 | 61.8 | 67.6 | 49.7 |
| STAR-2 / Q.+Size | 74M | 64MB | 7.2 | 28.7 | 37.5 | 52.8 | 61.0 | 68.9 | 49.8 |
| STAR-3 / Q.+Size | 70M | 151MB | 7.2 | 29.2 | 39.5 | 51.9 | 61.5 | 69.4 | 50.3 |
| STAR-4 / Q.+Size | 70M | 114MB | 7.2 | 29.2 | 40.0 | 52.7 | 61.4 | 68.9 | **50.4** |
| STAR-1 / Q.+Cache | 77M | 16MB | 7.2 | 28.9 | 40.0 | 51.3 | 61.0 | 66.4 | 49.5 |
| STAR-2 / Q.+Cache | 83M | 22MB | 7.2 | 28.7 | 40.1 | 50.3 | 62.2 | 66.0 | 49.5 |
| STAR-3 / Q.+Cache | 75M | 23MB | 7.2 | 28.9 | 40.6 | 50.2 | 61.3 | 67.2 | 49.6 |
| STAR-4 / Q.+Cache | 78M | 22MB | 7.2 | 29.1 | 39.9 | 53.0 | 62.2 | 66.7 | **50.2** |

Table 5.1: Evaluation of backbones optimized for quality (upper third), quality and size (middle third), and quality and cache (lower third). We test on LM-Eval-Harness (Gao et al., 2024), reporting Transformer++ and StripedMamba baselines trained on the same data. Size indicates trainable parameter count w/o embeddings.

| Backbone | Size | Cache (bytes \| 4K) | RedPj. ppl ↓ | ARC-c acc. norm. ↑ | Hella. acc norm. ↑ | ARC-e acc. ↑ | Wino. acc. ↑ | PiQA acc. ↑ | SciQ acc. ↑ | Avg. ↑ |
|---|---|---|---|---|---|---|---|---|---|---|
| Transf.++ | 1.2B | 805MB | 5.9 | 27.3 | 49.3 | 58.9 | 51.3 | 71.2 | 86.3 | 57.4 |
| StripedMb. | 1.1B | 136MB | 5.7 | 28.3 | 52.8 | 59.8 | 54.1 | 72.9 | 86.0 | 59.0 |
| STAR-1B | 1.1B | 86MB | 5.7 | 27.9 | 52.6 | 60.8 | 53.9 | 71.8 | 87.0 | 59.0 |

Table 5.2: Evaluation of a 1B STAR backbone (48 LIVs, 2048 width) optimized for quality and cache (LM-Eval-Harness (Gao et al., 2024)). Results are compared to parameter-matched Transformer++ and Striped-Mamba baselines trained on 40B RedPajama tokens. Size excludes embedding layers.

## 5.3 SYNTHESIZING PARAMETER-EFFICIENT HIGH-QUALITY LANGUAGE MODELS

We observed that STAR can synthesize high-quality language models. Next, we ask whether it can likewise synthesize language models of high quality and smaller parameter counts. To this end, we evolve a population using evaluation perplexity and parameter count as objectives. Optimizing for quality and size, STAR evolution improves an initial population by 0.5 PPL points at an average reduction of 10% in trainable parameter count, as shown in Figure 1.1. We also evaluate the performance of representative synthesized backbones when training them longer.

> **Finding 3:** STAR backbones–optimized for quality and size–outperform Transformer++ and match StripedMamba backbones in RedPajama eval PPL, while surpassing both on Hellaswag, ARC-Easy, Winogrande, PiQA, and SciQ, with a reduction in parameter count by 13% and 8% respectively (Table 5.1 and A.4).

## 5.4 SYNTHESIZING CACHE-EFFICIENT HIGH-QUALITY LANGUAGE MODELS

High inference costs limit the widespread use of language models. To address this, we test whether STAR can synthesize architectures with reduced inference cache size without sacrificing predictive quality. By optimizing for perplexity and cache size (Fig. 5.3), STAR evolution improves an initial population by 0.4 PPL points and a 40% cache size reduction.

> **Finding 4:** STAR backbones–optimized for quality and cache size–outperform Transformer++ and match StripedMamba in RedPajama perplexity while surpassing both on HellaSwag, ARC-Easy, Winogrande, and PiQA, with cache size reductions of 90% and 36%, respectively, at a sequence length of 4096 tokens (Tables 5.1 and A.4).

Our previous experiments have shown that STAR synthesis at smaller scales yields suboptimal results (see Fig. 5.1). Nevertheless, when scaling a 24-LIV backbone at 768 dimensions, optimized for quality and cache size, to 48 LIVs at 2048 dimensions, through stacking and width extension,

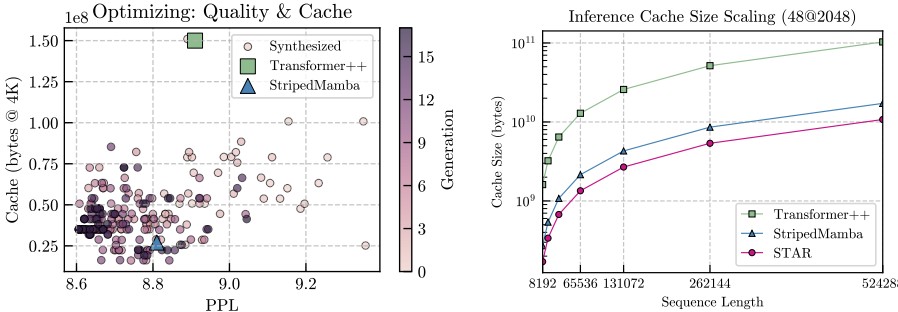

Figure 5.3: **[Left]:** Genome scores during `STAR` evolution when optimizing for quality and cache size. Cache size is computed at a fixed sequence length of 4096 tokens. **[Right]:** Cache size scaling with increasing input sequence length for the models show in Table 5.2.

we find that it matches the performance of a parameter-matched StripedMamba baseline and outperforms a parameter-matched Transformer++ baseline when all are trained for 40B tokens:

> **Finding 5:** Scaling a synthesized `STAR` backbone from 125M to 1B parameters (Fig. B.25) outperforms a parameter-matched Transformer++ baseline and matches a StripedMamba baseline in RedPajama evaluation perplexity and performance on ARC-Challenge, HellaSwag, Winogrande, PiQA, and SciQ, while reducing cache size by 90% and 37%, respectively (Table 5.2).

> **Finding 6:** Overall, synthesized `STAR` backbones outperform Transformer++ and StripedMamba baselines with hit rates of 8/8, 7/8, and 7/8 when optimizing for quality, quality and size, and quality and cache, respectively (Tables 5.1 and A.4).

### 5.5 COMPARING SYNTHESIZED BACKBONES

`STAR` is well-suited for the synthesis of backbones optimized for various objectives. In addition, it provides a tool for the automated discovery of backbone motifs that drive these performance improvements, as it evolves populations towards using those combinations and compositions of LIVs that perform best. Figure 5.4 demonstrates this for the evolution targeting model quality and size. We observe that `STAR` favors gated short convolutions (GConv-1), grouped query (Ainslie et al., 2023) attention variants (SA-3), and differential variants of input-varying recurrences (Rec-1-Diff), as well as SwiGLUs (Shazeer, 2020) (GMemless). A more detailed analysis of the motifs resulting from all `STAR` evolutions, as well as visualizations of all evaluated backbones, can be found in Appendix B.

## 6 CONCLUSION

This work presents `STAR`, a framework for the automated evolution of tailored architectures. Unlike other approaches, `STAR` explores a hierarchical, general design space encompassing attention, recurrences, convolutions, and other input-dependent units. Its design space is well-conditioned, with most architecture candidates training stably. Using evolutionary methods on a numerical backbone encoding, `STAR` achieves significant improvements in perplexity, downstream benchmarks, model size, and inference cache compared to optimized striped hybrid and Transformer baselines.

**Reproducibility statement** To aid in reproducibility, we run optimization and training on open-source datasets (RedPajama). We report full training and `STAR` evolutionary algorithms details in Appendix A and a full description of the `STAR` genome in Appendix A.6.

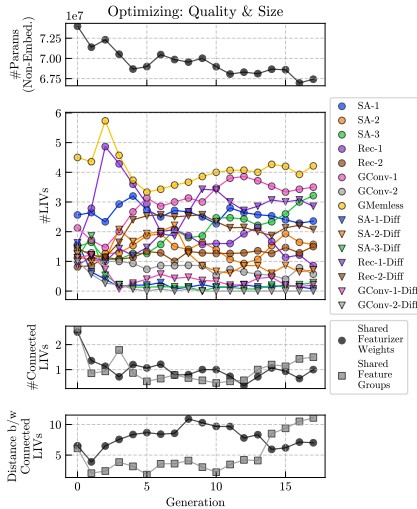

Figure 5.4: Evolution of backbones optimized for quality and size, averaged per population. Distance measures the number of other LIVs between two connected LIVs.

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

# Supplementary Material

CONTENTS

# A    EXPERIMENTAL DETAILS

## A.1    DISCUSSION

**Limitations and extensions of** STAR    While the LIV search space is general in the context of sequence modeling primitives, it does not include all classes of functions that can be embedded in a backbone, potentially missing options for further optimization.

**Relation to scaling laws and mechanistic design**    Since STAR evolution can target any objective, the methods presented in this paper are suited to integration within scaling laws protocols. Another options is optimizing efficiently computed metrics that correlate with performance at scale e.g., average accuracy on curated synthetic tasks (Arora et al., 2023; Poli et al., 2024).

**Optimizing variable depth backbones**    Currently, STAR optimizes fixed-length genomes, limiting architectures to fixed depth and width. Optimizing variable-depth and variable-width architectures is challenging due to the hierarchical and modular design space. Shallower genomes are computationally cheaper and converge faster but may lack the complexity needed for difficult tasks. Deeper genomes offer greater representational power but expand the search space, risking suboptimal convergence due to overfitting or inefficient sampling. Future extensions of STAR could address these challenges with adaptive mechanisms like depth-aware sampling or dynamic penalties to improve scalability. Testing these methods on tasks requiring deeper architectures will be key to enhancing STAR's robustness and versatility.

**Multi-level optimization**    STAR streamlines the search by treating the genome as a unified entity, enabling efficient exploration—especially beneficial for rapid iteration or limited resources. However, this approach may not fully exploit the hierarchical design space, potentially overlooking dependencies or key subspaces across genome levels. In contrast, a multi-stage optimization strategy could systematically refine each hierarchical level—featurization, operator structure, and backbone—using tailored evolutionary algorithms. This could improve convergence, especially in complex task settings, by leveraging the genome's modularity and addressing interactions incrementally, but it would increase algorithmic complexity and computational costs.

## A.2    TRAINING

Tables A.1, A.2, and A.3 provide an overview of the training settings used during STAR evolution and when evaluating the resulting synthesized backbones.

Table A.1: Training setting during STAR evolution.

| OPTIMIZER | ADAMW |
|---|---|
| OPTIMIZER MOMENTUM | $\beta_1, \beta_2 = 0.9, 0.95$ |
| BATCH SIZE | 0.25M TOKENS |
| TRAINING STEPS | 5000 |
| LEARNING RATE SCHEDULE | COSINE DECAY |
| LINEAR LEARNING RATE WARM-UP | 500 STEPS |
| BASE LEARNING RATE | 0.0008 |
| WEIGHT DECAY | 0.1 |
| DROPOUT | NONE |
| GRADIENT CLIPPING | 1.0 |

Table A.2: Training setting for evaluation of synthesized backbones.

| OPTIMIZER | ADAMW |
|---|---|
| OPTIMIZER MOMENTUM | $\beta_1, \beta_2 = 0.9, 0.95$ |
| BATCH SIZE | 0.25M TOKENS |
| TRAINING STEPS | 20000 |
| LEARNING RATE SCHEDULE | COSINE DECAY |
| LINEAR LEARNING RATE WARM-UP | 1500 STEPS |
| BASE LEARNING RATE | 0.0008 |
| WEIGHT DECAY | 0.1 |
| DROPOUT | NONE |
| GRADIENT CLIPPING | 1.0 |

Table A.3: Training setting for models with 48 LIVs at a width of 2048 dimensions (see Table 5.2).

| OPTIMIZER | ADAMW |
|---|---|
| OPTIMIZER MOMENTUM | $\beta_1, \beta_2 = 0.9, 0.95$ |
| BATCH SIZE | 0.75M TOKENS |
| TRAINING STEPS | 50000 |
| LEARNING RATE SCHEDULE | COSINE DECAY |
| LINEAR LEARNING RATE WARM-UP | 3500 STEPS |
| BASE LEARNING RATE | 0.0008 |
| WEIGHT DECAY | 0.1 |
| DROPOUT | NONE |
| GRADIENT CLIPPING | 1.0 |

### A.3 BASELINE MODELS

All baseline models are trained according to the recipe described in Table A.2. We train the two baseline models each at two depths widths to match the parameter counts of our synthesized models (Tables 5.1 and 5.2): 24 operators at 768 dimensions and 48 operators at 2048 dimensions.

**Transformer++**  A Transformer (Vaswani et al., 2017) with an improved architecture, namely rotary positional encodings (Su et al., 2024), SwiGLU MLP (Shazeer, 2020), RMSNorm instead of LayerNorm, and no linear bias term. We use a head dimension of 64 for all Transformer++ baselines trained in this work, resulting in 12 and 32 heads for models with 768 and 2048 width.

**StripedMamba**  A striped hybrid backbone (Poli et al., 2024) that combines Mamba (Gu & Dao, 2023), SwiGLU MLP (Shazeer, 2020), and self-attention (Vaswani et al., 2017) operators. The 24 operator backbone is composed of interleaved Mamba and SwiGLU operators, with the exception of operators 6 and 18, which are softmax attention. The Mamba operators have a state size of 32, while we use a head dimension of 64 for the attention operators. The 48 operator StripedMamba backbone is obtained by stacking two 24 operator backbones in depth and increasing the overall width to 2048.

### A.4 EVOLUTIONARY OPTIMIZATION ALGORITHMS

In this section, we present an overview of the three variants of commonly used gradient-free evolutionary optimization algorithms applied in this work. These algorithms have been adapted as necessary to be compatible with the STAR genome. Before discussing their individual differences, we will first describe several core operations shared across all variants.

**Tournament selection**  selects parent candidates by randomly sampling a subset of individuals from the population and choosing the highest-performing individual from this subset. This method promotes the propagation of strong candidates while preserving diversity through its inherent randomness.

**K-point crossover**  recombines the genomes of two parents by exchanging segments of genetic material at k randomly selected points, creating offspring that inherit a mix of traits from both parents.

**Elitism**  balances exploration and exploitation by preserving a subset of the top-performing individuals from the current population and carrying them over directly to the next generation. This approach ensures that high-quality solutions are not lost and generally accelerates convergence, while reducing the risk of premature convergence to suboptimal regions of the solution space.

**Mutation**  maintains population diversity by introducing randomness through random alterations to a genome, helping the algorithm explore new regions of the solution space.

**Firefly Algorithm (FA)**  The Firefly Algorithm (FA) is inspired by fireflies' attraction to brighter (fitter) individuals based on their light intensity. FA assigns a light intensity $a = \frac{1}{1+s}$ to each genome, inversely related to its fitness score $s$. In each iteration, FA pairs genome $i$ with genome $j$ via tournament selection. If $a_j > a_i$, FA updates $g^i$ to resemble $g^j$ through two steps: (1) computing attraction strength $\beta = \beta_0(1 - e^{-\gamma(1-r)})$, where $\beta_0$ is baseline attraction, $\gamma$ is the light absorption coefficient, and $r$ is the similarity ratio of matching LIVs, and (2) replacing LIV $g_k^i$ with $g_k^j$ with probability $\beta$. If $a_i \geq a_j$, $g^i$ remains unchanged. Finally, $g^i$ undergoes mutation.

**Genetic Algorithm (GA)**  In each iteration, the Genetic Algorithm (GA) generates new genomes by: (i) selecting two parents via tournament selection; (ii) recombining them using k-point crossover; and (iii) mutating the recombined genomes.

**Non-dominated Sorting Genetic Algorithm II (NSGA-2)**  NSGA-2 extends GA for multi-objective optimization by maintaining a diverse set of Pareto-optimal solutions through non-dominated sorting and crowding distances. It first segregates genomes into fronts, with the first front containing the most optimal, non-dominated solutions. Genome $g^i$ dominates $g^j$ if it outperforms $g^j$ in at least one objective without being worse in others. Within each front, genomes are sorted by objective scores, such as predictive quality, and assigned crowding distances [9] Boundary solutions, with extreme objective scores, receive infinite crowding distances to ensure preservation, while non-boundary solutions are assigned crowding distances based on differences from adjacent neighbors. Selection then favors genomes with higher front rank and crowding distance.

**Determining hyper-parameters for NSGA-2**  We found that NSGA-2 performed the best in our comparison (Section 5). We also investigate the optimal hyper-parameter settings for NSGA-2, specifically population size $n$, mutation probability $p$, and number of crossover points $k$. To do this, we evolved two population sizes (16 and 32), optimizing for quality and parameter count, while varying the number of crossover points (1 or 2) and mutation probabilities (10% or 20%) (Fig. A.1). To keep to overall number of sampled genomes constant, we evolve populations containing 16 genomes for 8 iterations and populations containing 32 genomes for 4 iterations. All genomes contain 24 LIVs at a width of 64 dimensions. Our results indicate that NSGA-2 performs best with a population size of 16, 2 crossover points, and a 10% mutation probability.

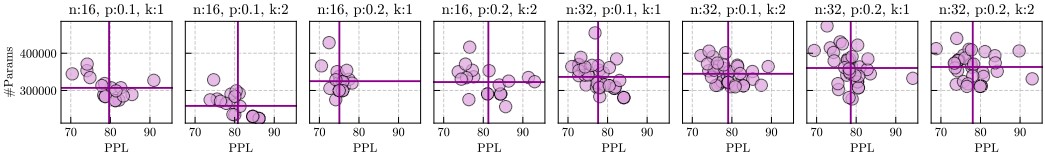

Figure A.1: Comparison of different hyper-parameter settings for NSGA-2.

## A.5 GENOME SCORES

We provide visualizations of all genome scores in all STAR evolutions:

i. **Evolutionary algorithm ablations**: figures B.28 and B.29.

ii. **Comparison of synthesis scales**: figures B.30, B.31, and B.32.

iii. **Direct quality optimization**: figure B.34.

iv. **Quality and size optimization**: figure B.33.

v. **Quality and cache optimization**: figure B.35.

## A.6 LINEAR INPUT-VARYING SYSTEMS AND FEATURIZERS: OPTION POOLS

Modifying the STAR genome – the numerical encoding enabled by the LIV framework – allows exploration of model architectures with substantial differences at multiple levels, including in the type of LIVs they are composed of, as well as their ordering and interconnection.

Recall that the STAR genome is structured hierarchically: At the highest level, the *backbone genome* specifies the composition of LIVs in the backbone, with each LIV represented by a single integer. Expanding this integer reveals the *operator genome*, which identifies the LIV. At the operator genome level, the specific featurizer of the LIV is similarly encoded as a single integer, which can

---

[9]Crowding distance in multi-objective optimization is a measure of the density of solutions surrounding a particular solution, calculated as the sum of the normalized distances between adjacent solutions across all objectives: $d_i = \sum_{m=1}^{M} \left( \frac{f_m^{i+1} - f_m^{i-1}}{f_m^{\max} - f_m^{\min}} \right)$, where $d_i$ is the crowding distance for solution $i$, and $f_m$ is the objective value in the $m^{th}$ objective.

be further expanded into the *featurizer genome*, which specifies the particular featurizer used by the LIV.

Below, we provide an overview of the specific integer values and their corresponding meanings considered in this work for each level of the STAR genome. Note that this definition of the genome is not exhaustive and can be extended further.

### A.6.1 BACKBONE GENOME

The basic formulation of the backbone genome (without extensions such as residual composition explored in A.7) consists of sequences of five integers, where each sequence corresponds to one of the LIVs contained in the backbone, defining (a) the individual operator and (b) composition rules with other LIVs.

For each integer in the genome, we detail the set of options considered in this work. While we present a specific set of choices here, the pool of options can be readily expanded, provided the results compile to valid operators and backbones within the LIV framework.

Tthe **first integer** specifies the LIV class and can take on the following values in our experiments:

- 1-4. Softmax attention variants (SA) 1-4
- 5-6. Recurrences (Rec) 1-2
- 7-8. Gated convolutions (GConv) 1-2
- 9. Gated memoryless unit (GMemless)
- 10-17. Differential variants of LIV classes 1-8 (akin to the "Differential Transformer" (Ye et al., 2024))

The **second integer** defines the weight-sharing structure of the LIVs' featurizers. Specifically, all LIVs within a backbone that share featurizer weights will have the same value in this position. The mapping of integer values to the weight-sharing structure thereby depends on the number of occurrences of each LIV class ($N$) in the backbone. If all LIVs of the same class share featurizer weights, they are all assigned a value of 1 at this position. Conversely, if none of these LIVs share featurizer weights, each LIV is assigned a unique integer value from 1 to $N$.

The **third integer** defines the strategy for sharing featurizer weights. In this work, we are considering only two possible featruizer sharing strategies:

1. No weights are shared
2. All weights are shared

The **fourth integer** establishes the feature group sharing structure of the LIVs. Similar to the featurizer weight-sharing structure (second integer), all LIVs within a backbone that share feature groups will have the same value at this position. The assignment of integer values to feature group sharing follows the same logic described for the second integer.

The **fifth integer** specifies the strategy used for sharing feature groups. Since feature groups are unique to each LIV class, the possible values for this integer vary depending on the LIV class. The range is between 1 (indicating no shared feature groups) and $N+1$, where $N$ represents the number of unique feature groups in the given LIV class. For example, in the case of softmax attention, the possible values are:

1. No shared feature groups
2. Shared key cache
3. Shared value cache
4. Shared key and value cache

For clarity, we provide **examples of backbone genomes** below:

> 11111   91111   12121   92121

This genome consists of four LIVs arranged in an interleaved order. The first and last LIVs belong to the SA-1 class, while the second and fourth LIVs belong to the GMemless class. None of the LIVs share featurizer weights or feature groups, as each occurrence of a LIV class has distinct integer values for featurizer sharing (integer 2) and feature group sharing (integer 4). Both the featurizer sharing strategy (integer 3) and feature group sharing strategy (integer 5) are set to 1, indicating no sharing.

> 11111   91111   51111   92121

This genome represents a variation of the genome shown above, where the third LIV has been switched from class SA-1 to class Rec-1. As before, none of the LIVs share feature groups of featurizer weights.

> 11111   91111   51111   92121   11221   91131

This genome is comprised of six LIVs. The first and fifth belong to the SA-1 class. The first and fifth belong to the SA-1 class, the second, fourth, and sixth to the GMemless class, and the third to the Rec-1 class. Notably, the two SA-1 LIVs share the weights of their featurizers, as both have a value of 1 at the second integer position and a value of 2 at the third integer position.

### A.6.2   OPERATOR GENOME

The operator genome specifies a particular LIV and consists of five integer values. Integer 1 summarizes the LIV's featurizer, integers 2–4 define the token-mixing structure of the LIV, and integer 5 determines the channel-mixing structure.

The **first integer** specifies the featurizer class. In this work, we consider the following 9 featurizer classes:

1. Dense channel mixing structure with diagonal token mixing structure on all feature groups (3 groups e.g., in SA-1).

2. Dense channel mixing structure with Toeplitz token mixing structure on all feature groups (3 groups e.g., in SA-2).

3. Variant of 1. where a repeat factor of 4 is applied to the last two feature groups (e.g., used for SA-3).

4. Variant of 1. where a repeat factor of 2 is applied to the last two feature groups (e.g., used for SA-4).

5. Dense channel mixing and Toeplitz token mixing structure. An expansion factor of 16 is applied to the last two feature groups (e.g., used for Rec-1).

6. Dense channel mixing and Toeplitz token mixing structure. An expansion factor of 2 is applied to the last two feature groups (e.g., used for Rec-2).

7. Diagonal channel mixing structure with Toeplitz token mixing structure for all feature groups. One of the groups is explicitly parametrized (e.g., short convolutions of length 3 used for GConv-1).

8. Variant of 5. Where the short convolution kernel feature group is replaced with an implicitly parametrized feature group (e.g., long convolutions used in GConv-2).

9. Dense channel mixing structure with diagonal token mixing structure with 2 feature groups (e.g., used for GMemless). Variant of 1 with one fewer feature group.

The **second integer** defines the linear token-mixing structure of the LIV, before any final nonlinearity, and can take on the following values in this work:

1. Diagonal (e.g., GMemless)

2. Low rank (e.g., SA)

3. Scaled Toeplitz (e.g., GConv)

4. Sequentially semi-separable (e.g., Rec)

The token-mixing structure determines the class of matrix multiplication algorithms that can be used to apply the operator to the input. For instance, if the LIV is sequentially semi-separable, it supports an $O(l)$ algorithm implemented as a linear recurrence.

The **third** integer determines whether any structured sparsity mask is applied to the token-mixing structure. We consider the following in this work:

1. No sparsity

2. Banded (e.g., as used for short convolutions)

Note that all models trained in this work are causal, and as such upper-triangular sparsity masks are introduced whenever needed e.g., in LIVs wrapped by nonlinearities.

The **fourth integer** describes whether any final nonlinearity is applied to the token-mixing structure. We consider the following set of static and normalization non-linearities in this work:

1. None

2. Softmax

3. ReLU

4. Swish

The **fifth integer** describes the LIV channel mixing structure, for which we consider the following two possible structures in this work:

1. Diagonal

2. Dense

Below we provide the specific operator genomes for each LIV class considered in this work:

SA-1 `12121` refers to the standard attention operator using a featurizer with a dense channel mixing and diagonal token mixing structure with 3 feature groups, a low-rank token-mixing structure, no sparsity mask, a softmax non-linearity, and a diagonal channel-mixing structure.

SA-2 `22121` represents a variant of SA-1 whose featurizer has a dense channel mixing and Toeplitz token mixing structure. This is realized in practice by adding depth-wise convolutions to the featurizer, in line with the findings of (So et al., 2021; Poli et al., 2023).

SA-3 `32121` represents a variant of SA-1 where a repeat factor of 4 is applied to the last two feature groups. This is akin to variants of *multi-query* (MQA) and *grouped-query attention* (GQA) (Shazeer, 2019).

SA-4 `42121` represents a variant of SA-3 with a lower repeat factor of 2 for the last two feature groups.

Rec-1 `54111` is characterized by a featurizer with Toeplitz token mixing structure and dense channel mixing structure with 5 feature groups where an expansion factor of 16 is applied to the last two feature groups, a semi-separable token mixing structure, no sparsity, no non-linearity, and a diagonal channel mixing structure. Rec-1 is representative of a variety of modern input-varying linear recurrent layers (Gu & Dao, 2023; Yang et al., 2024a).

Rec-2 `64111` is the same as Rec-1, with the exception of an expansion factor of 2 for the last two feature groups.

GConv-1 `73111` is characterized by a featurizer with diagonal channel mixing structure and Toeplitz token mixing structure (using a short convolution filter) applied to all feature groups in

addition to an explicitly parametrized feature group for short convolution, a Toeplitz token mixing structure, no sparsity, no non-linearity, and a diagonal channel mixing structure.

GConv-2 `83111` has the same structure as GConv-1, except for the use of an implicitly parametrized feature group for long convolutions. GConv-2 represents modern operators in the gated *long convolution* family (Poli et al., 2023).

GMemless `91142` is characterized by a featurizer with dense channel mixing structure and diagonal token mixing structure with two feature groups, a diagonal token mixing structure, no sparsity, swish non-linearity, and dense channel mixing structure. GConv-1 thereby represents a SwiGLU (Shazeer, 2020).

In addition, we include differential variants of LIV classes 1–8 (SA, Rec, and GConv), where two identical LIVs are applied in parallel, outputting their difference, similar to the "Differential Transformer" (Ye et al., 2024).

### A.6.3 FEATURIZER GENOME

The featurizer genome is composed of sequences of seven integers, one sequence per feature group of the featurizer. The **first five integers** are akin to integers 2-5 of the operator genome, respectively indicating the linear token-mixing structure, whether any sparsity is applied, whether any non-linearity is applied, and the channel mixing structure. The **sixth integer** indicates an expansion factor of the feature group channel dimension over the input channel dimension. The **seventh integer** indicates a repeat factor for how many times the feature groups are replicated across the channel dimension.

Note that we restrict the featurizer genome to a maximum of 5 feature groups (ie, 35 integers in total). If the featurizer takes in less than 5 feature groups, we set the sequences of all excess feature groups to 0.

### A.7 EXTENDING THE BACKBONE GENOME FOR VARIABLE RESIDUAL CONNECTIONS

In our main experiments, we constrain backbone topologies to a pre-norm residual structure, where the output $y^m$ of LIV $T^m$ at backbone depth $m$ is defined as: $y^m = T(\text{norm}(y^{m-1}))\,\text{norm}(y^{m-1}) + y^{m-1}$.

However, the backbone genome can be extended to support more flexible residual streams. This can be achieved by introducing a sixth entry to each subsection of the backbone genome, corresponding to its respective LIV. Recall that the backbone genome consists of sequences of five integers, where each sequence encodes the characteristics of an LIV and its integration within the composition structure. If two LIVs, at depths $m$ and $n$ (where $m < n$), share the same value at this new genome position, the residual stream is extended such that: $y^n = T(\text{norm}(u))\,\text{norm}(u) + u,\quad \text{where } u = y^{n-1} + y^m$.

We evaluate this extended backbone genome in an ablation study by comparing the outcomes of two `STAR` evolutions: one incorporating this extension and the other using the standard backbone genome. In both conditions, we evolve a population of 16 genomes, each consisting of 24 LIVs with a width of 768 dimensions, for 7 generations. The results indicate that the extension allows STAR to synthesize architectures of even smaller parameter counts while maintaining the same level of quality (see Fig. A.2).

Based on these promising findings, we plan to further investigate composition strategies via residuals and inputs to LIVs in future work, in addition to improved featurizer interconnections e.g., sharing inputs to the system, the featurizer, and the residual stream itself.

### A.8 EVALUATION OF SYNTHESIZED BACKBONES

Table A.4 provides an overview of the evaluation performances of the remaining 4 synthesized backbones that were selected from each `STAR` evaluation and trained for longer (for comparison, see Table 5.1).

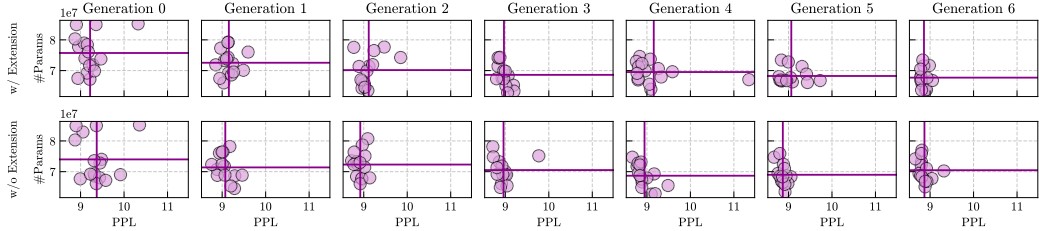

Figure A.2: Comparison of two `STAR` evolutions with and without an extension of the backbone genome allowing for more flexible residual connections.

| Backbone / Optimized for | Size | Cache (bytes \| 4K) | RedPj. ppl ↓ | Hella. acc. norm. ↑ | ARC-e acc. ↑ | Wino. acc. ↑ | PiQA acc. ↑ | SciQ acc. ↑ | Avg. ↑ |
|---|---|---|---|---|---|---|---|---|---|
| Transformer++ | 85M | 150MB | 7.3 | 28.9 | 38.8 | 51.2 | 61.2 | 64.1 | 48.8 |
| StripedMamba | 80M | 25MB | 7.2 | 28.6 | 39.3 | 51.1 | 60.9 | 67.4 | 49.5 |
| `STAR`-5 / Quality | 78M | 94MB | 7.1 | 29.2 | 39.1 | 52.1 | 62.1 | 72.7 | **51.0** |
| `STAR`-6 / Quality | 79M | 94MB | 7.1 | 29.0 | 39.9 | 50.9 | 61.7 | 71.1 | 50.5 |
| `STAR`-7 / Quality | 79M | 107MB | 7.1 | 29.3 | 38.2 | 51.5 | 61.6 | 70.2 | 50.2 |
| `STAR`-8 / Quality | 79M | 94MB | 7.1 | 29.1 | 40.6 | 50.8 | 62.0 | 70.3 | 50.6 |
| `STAR`-5 / Q.+Size | 78M | 64MB | 7.2 | 29.2 | 40.0 | 52.7 | 61.0 | 67.8 | 50.1 |
| `STAR`-6 / Q.+Size | 73M | 104MB | 7.2 | 27.7 | 39.5 | 53.1 | 61.6 | 69.4 | **50.3** |
| `STAR`-7 / Q.+Size | 69M | 170MB | 7.2 | 27.8 | 39.2 | 49.9 | 61.2 | 69.5 | 49.5 |
| `STAR`-8 / Q.+Size | 72M | 92MB | 7.2 | 27.5 | 39.2 | 51.7 | 61.8 | 64.1 | 48.9 |
| `STAR`-5 / Q.+Cache | 79M | 22MB | 7.2 | 28.9 | 40.0 | 50.2 | 61.1 | 69.1 | 49.9 |
| `STAR`-6 / Q.+Cache | 68M | 25MB | 7.2 | 29.1 | 40.0 | 51.3 | 60.9 | 68.7 | **50.0** |
| `STAR`-7 / Q.+Cache | 75M | 22MB | 7.3 | 28.6 | 39.4 | 52.6 | 61.0 | 66.6 | 49.6 |
| `STAR`-8 / Q.+Cache | 74M | 16MB | 7.3 | 28.8 | 38.8 | 51.2 | 61.0 | 67.0 | 49.4 |

Table A.4: Evaluation of backbones optimized for quality (lower half) and quality and size (upper half). We test on LM-Eval-Harness (Gao et al., 2024), reporting parameter-matched Transformer++ and StripedMamba baselines trained on the same data. Size indicates trainable parameter count, excluding embeddings layers. All models were trained for 5B tokens from Redpajama.

# B   VISUALIZATION AND ANALYSIS OF STAR BACKBONES

We provide visualization of the `STAR` backbones presented in Tables 5.1, 5.2, and A.4. Featurizer sharing between operators is indicated as solid black arrows on the right, feature group sharing as dashed black arrows on the left.

   i. **Direct quality optimization**: figures B.1 (`STAR`-1), B.2 (`STAR`-2), B.3 (`STAR`-3), B.4 (`STAR`-4), B.5 (`STAR`-5), B.6 (`STAR`-6), B.7 (`STAR`-7), B.8 (`STAR`-8)

   ii. **Quality and size optimization**: figures B.9 (`STAR`-1), B.10 (`STAR`-2), B.11 (`STAR`-3), B.12 (`STAR`-4), B.13 (`STAR`-5), B.14 (`STAR`-6), B.15 (`STAR`-7), B.16 (`STAR`-8)

   iii. **Quality and cache optimization**: figures B.17 (`STAR`-1), B.18 (`STAR`-2), B.19 (`STAR`-3), B.20 (`STAR`-4),B.21 (`STAR`-5), B.22 (`STAR`-6), B.23 (`STAR`-7), B.24 (`STAR`-8), and B.25 (`STAR`-1B)

We also provide overviews of the average count at which each LIV type occurs in a population over the course of STAR optimization, the average count of LIVs that share featurizer weights or groups, and the average distance between LIVs sharing featurizer weights or feature groups[10]:

   i. **Direct quality optimization**: figure B.26

---

[10]Distance is indicated by the number of LIVs between two LIVs with connected through featurizer or feature group sharing.

ii. **Quality and size optimization**: figure 5.4

iii. **Quality and cache optimization**: figure B.27

We have observed that STAR can evolve populations of architectures to optimize their quality (perplexity, accuracy, downstream performance), size (number of parameters), and efficiency (inference cache). The basis for this is laid out by the flexibility of the LIV design space, which allows constructing computational units tailored to these various objectives. STAR leverages evolutionary optimization methods to search the design space and converge on those solutions performing best under the given set of objectives.

For example, a key mechanism for STAR to reduce parameter counts is to identify which LIVs can be connected through featurizer or feature group sharing without degrading performance. Likewise, STAR can reduce parameter counts by purposefully placing MLPs only at those positions of the backbone (as observed in Figs. B.9, B.10, B.11, B.12), instead of at every other depth index as is otherwise common.

By contrast, STAR can reduce cache size by deliberately placing LIVs with large cache sizes in the backbone and connecting these through featurizer or feature group sharing, while increasing the overall amount of MLPs in the backbone (as observed in Figs. B.27, B.17, and B.23).

## B.1 RECURRING MOTIFS

**Feature group sharing in softmax attention**   When optimizing solely for quality, a notable pattern in STAR is that the first LIV in the model is typically a variant of softmax attention (SA), connected via feature group sharing to other SA-LIVs positioned toward the end of the model (Figs. B.1, B.3, B.5, and B.7).

**Dominance of softmax attention and memoryless LIVs when optimizing for quality**   Softmax attention and memoryless LIVs are foundational to the Transformer architecture. When optimizing for quality, STAR tends to favor these LIV classes (Fig. B.26). Their performance is further enhanced by strategically placed recurrences and differential variants of short gated convolutions (Figs. B.1, B.4, and B.5).

**Sparsely placed differential gated convolutions with featurizer sharing**   Backbones optimized for quality often include two differential variants of short gated convolutions connected through featurizer sharing (as illustrated in Figs. B.2, B.4, B.5, B.6, and B.8).

**Reduced connectivity when optimizing for quality and size**   Interestingly, backbones synthesized by STAR for both quality and size exhibit significantly fewer LIVs connected through featurizer and feature group sharing compared to those optimized for quality alone (compare Figs. B.26 and 5.4).

**Connected gated convolutions**   A recurring motif from the evolutionary process involves LIVs with a block-Toeplitz token-mixing structure (e.g., convolutions, gated convolutions). In these cases, earlier LIVs in the model are connected through feature group sharing to later LIVs (Figs. B.12 and B.22).

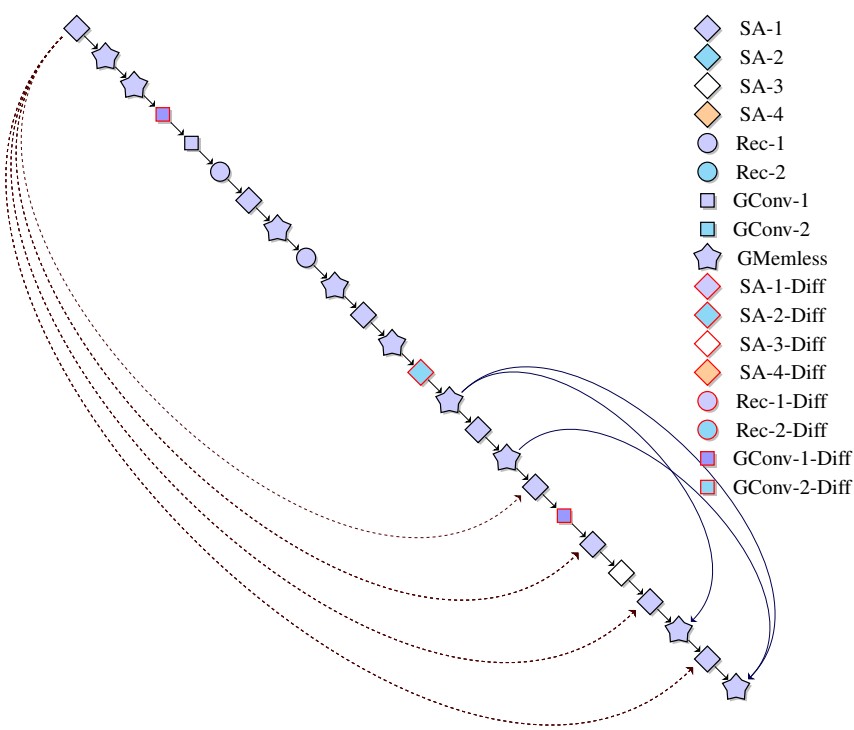

Figure B.1: STAR-1 optimised for quality (see Table 5.1). Dashed lines on the left indicate feature group sharing while solid lines on the right indicate featurizer sharing.

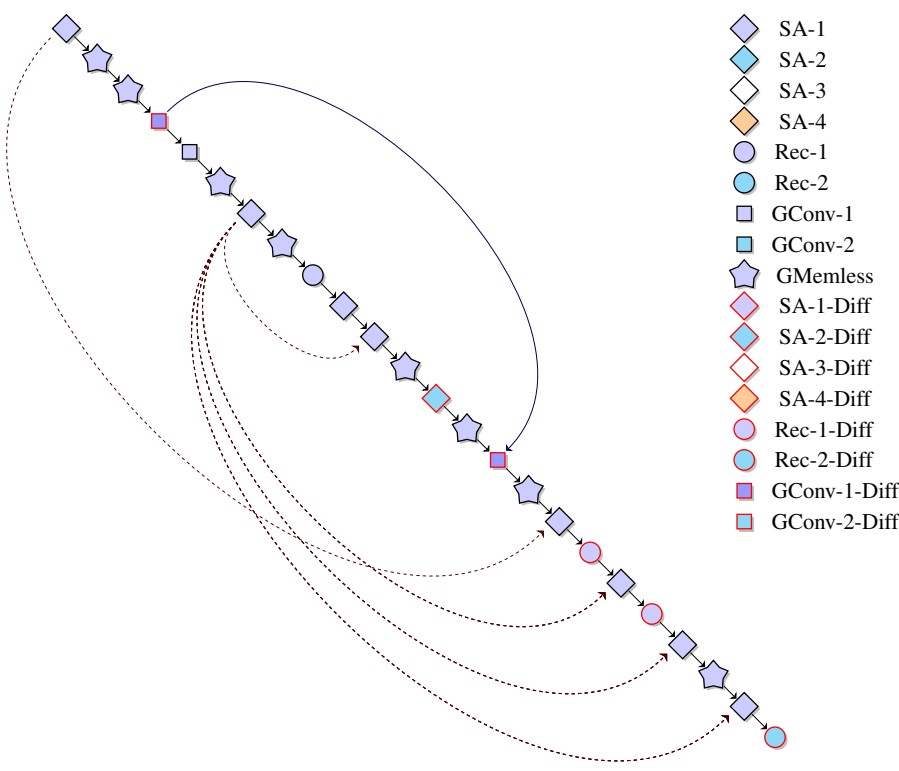

Figure B.2: STAR-2 optimised for quality (see Table 5.1). Dashed lines on the left indicate feature group sharing while solid lines on the right indicate featurizer sharing.

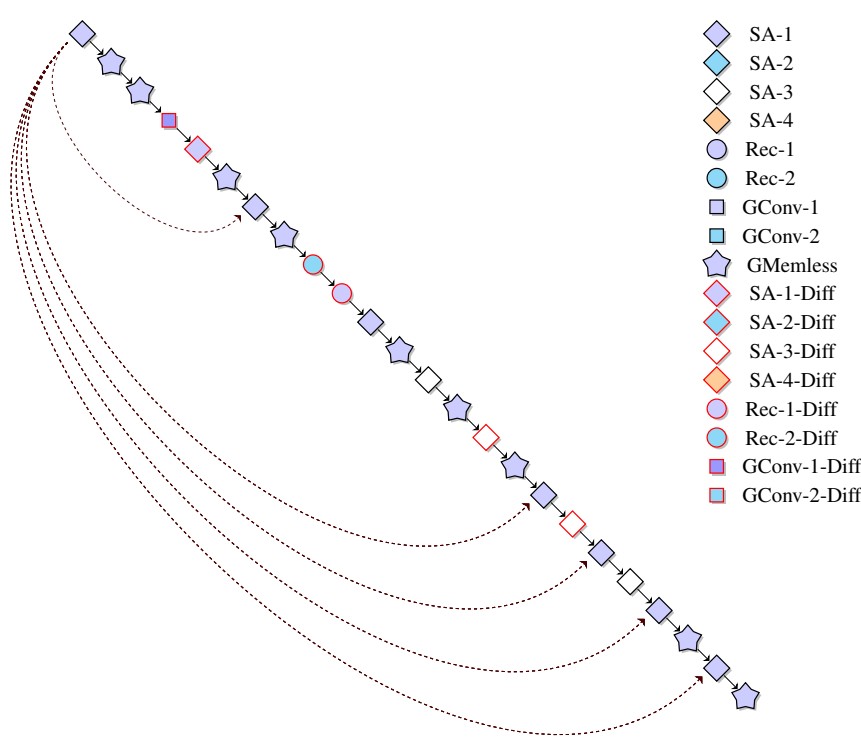

Figure B.3: STAR-3 optimised for quality (see Table 5.1). Dashed lines on the left indicate feature group sharing.

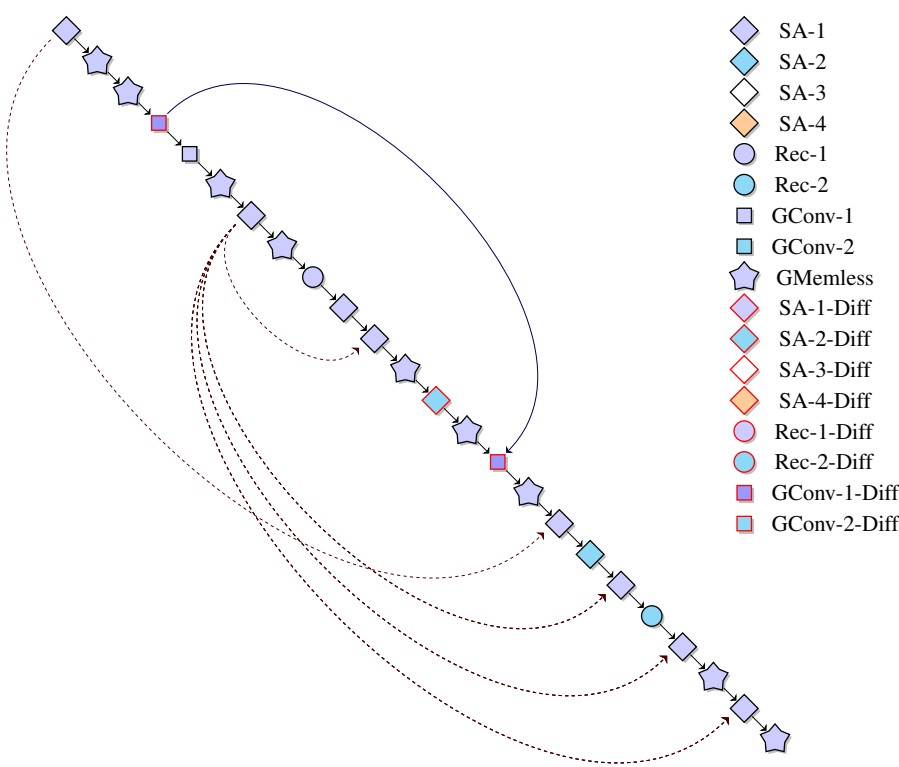

Figure B.4: STAR-4 optimised for quality (see Table 5.1). Dashed lines on the left indicate feature group sharing while solid lines on the right indicate featurizer sharing.

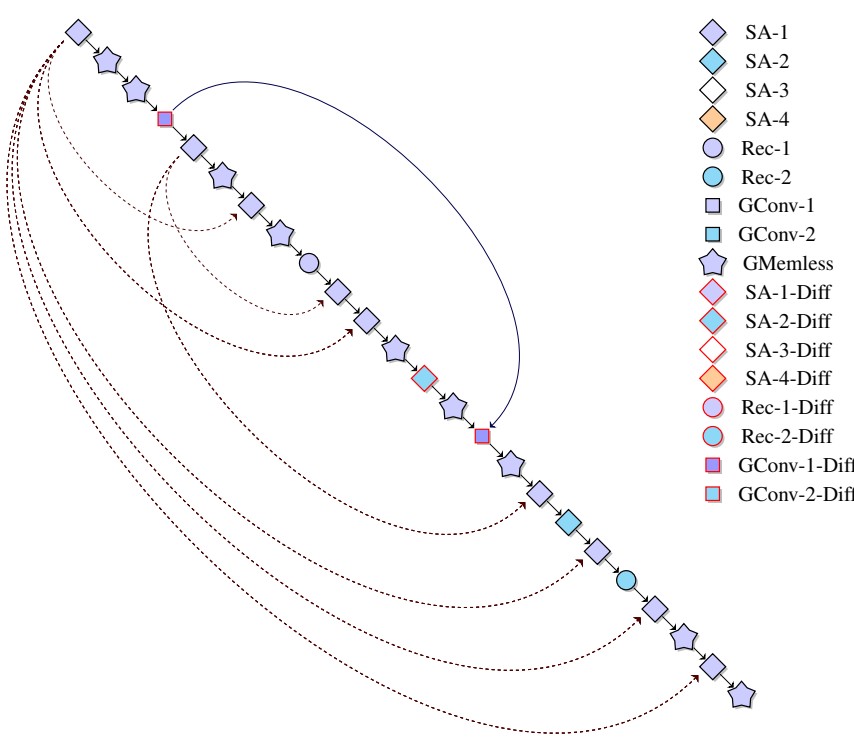

Figure B.5: STAR-5 optimised for quality (see Table A.4). Dashed lines on the left indicate feature group sharing while solid lines on the right indicate featurizer sharing.

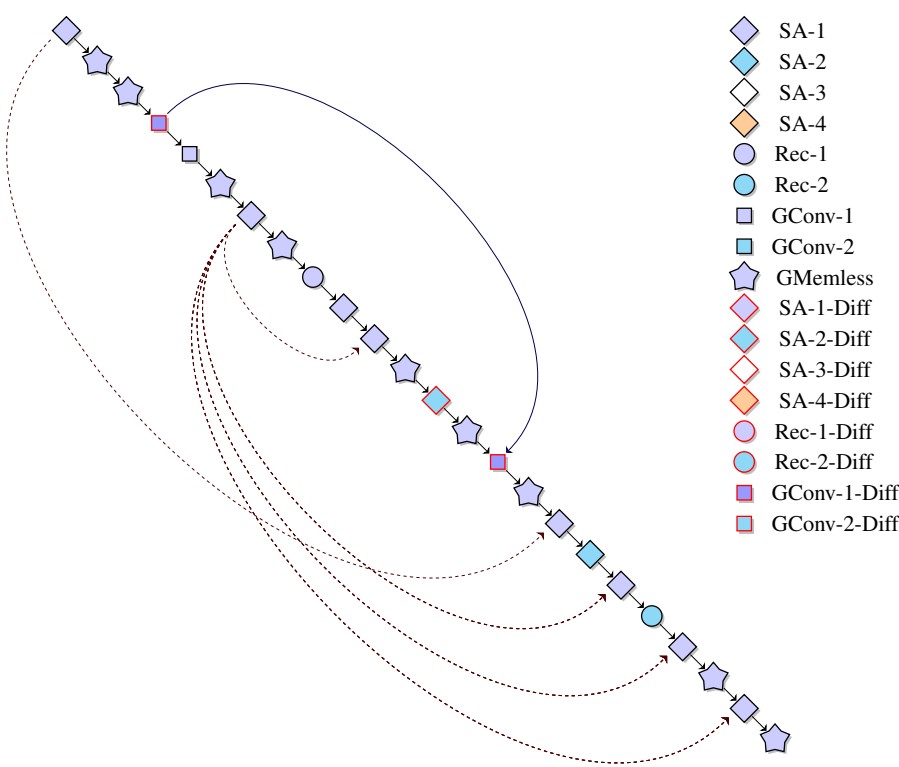

Figure B.6: STAR-6 optimised for quality (see Table A.4). Dashed lines on the left indicate feature group sharing while solid lines on the right indicate featurizer sharing.

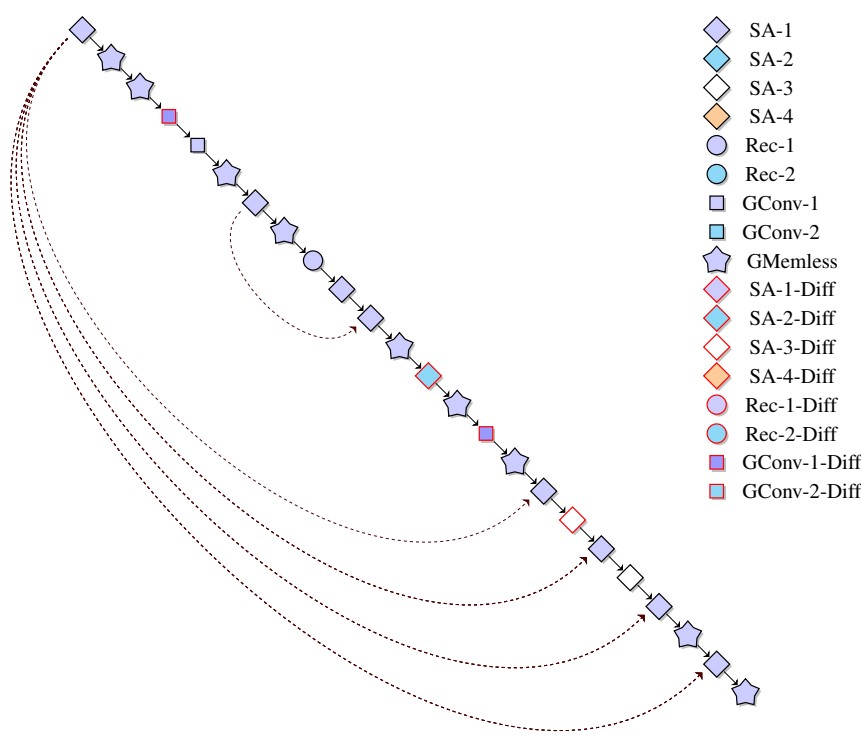

Figure B.7: STAR-7 optimised for quality (see Table A.4). Dashed lines on the left indicate feature group sharing.

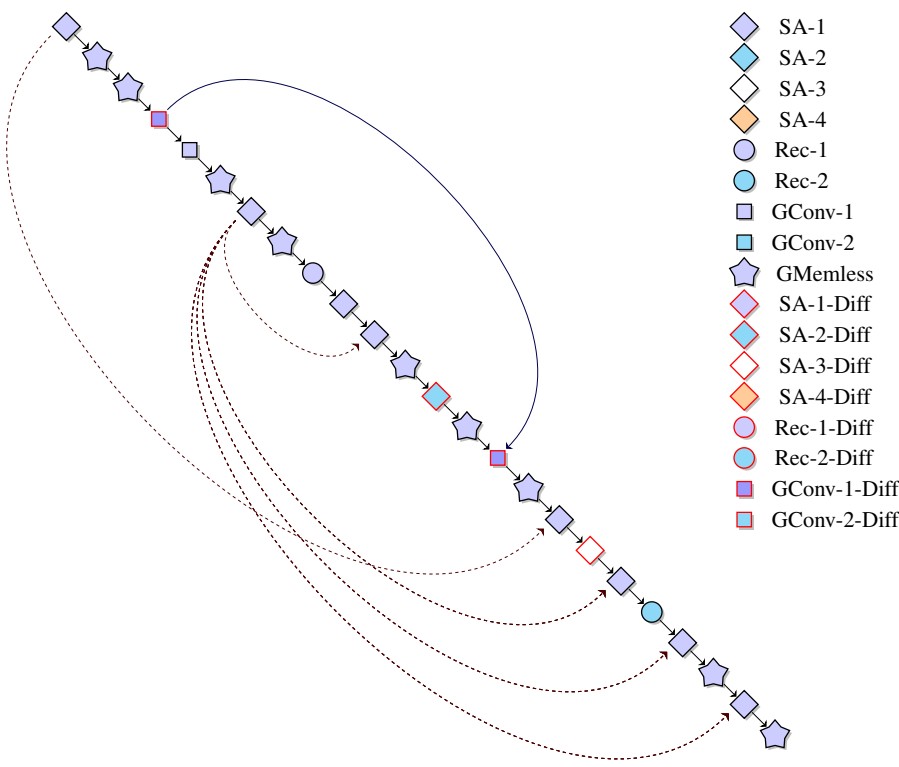

Figure B.8: STAR-8 optimised for quality (see Table A.4). Dashed lines on the left indicate feature group sharing while solid lines on the right indicate featurizer sharing.

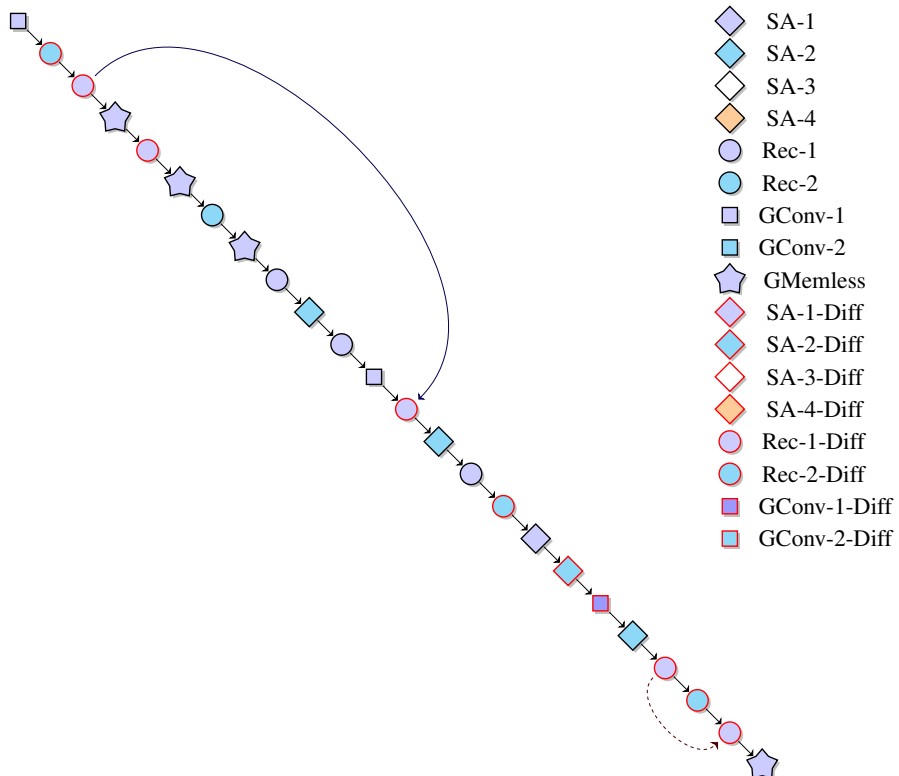

Figure B.9: STAR-1 optimised for quality and size (see Table 5.1). Dashed lines on the left indicate feature group sharing while solid lines on the right indicate featurizer sharing.

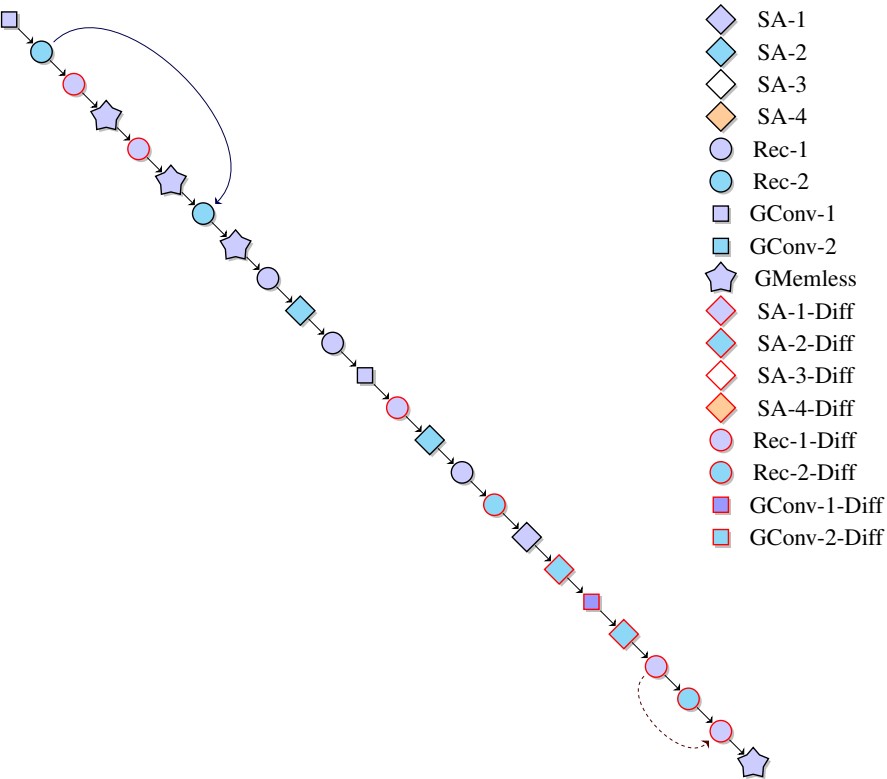

Figure B.10: STAR-2 optimised for quality and size (see Table 5.1). Dashed lines on the left indicate feature group sharing while solid lines on the right indicate featurizer sharing.

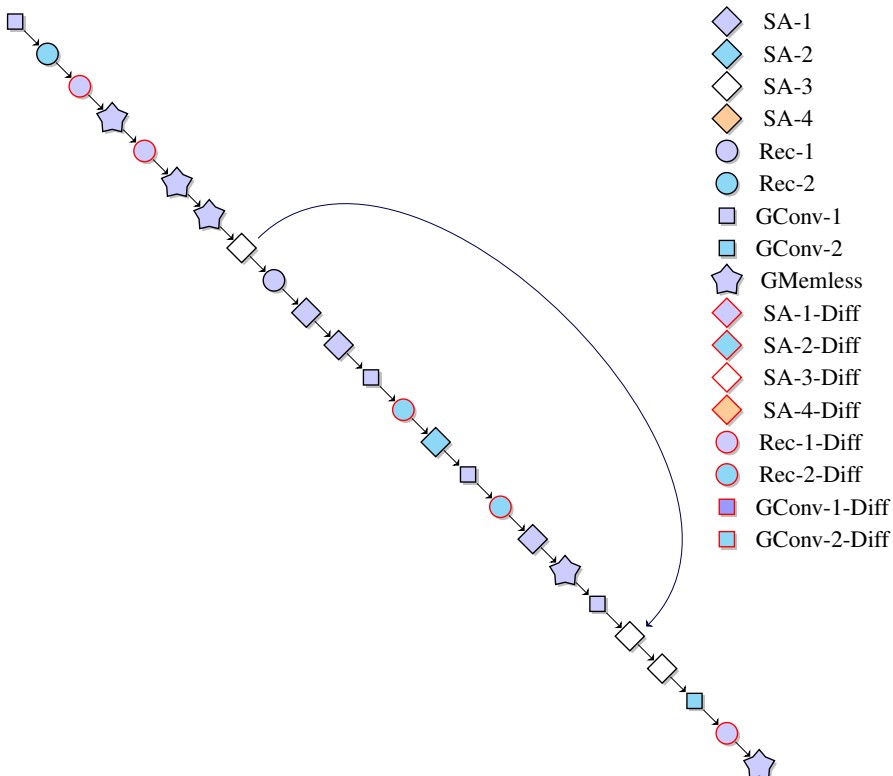

Figure B.11: STAR-3 optimised for quality and size (see Table 5.1). Solid lines on the right indicate featurizer sharing.

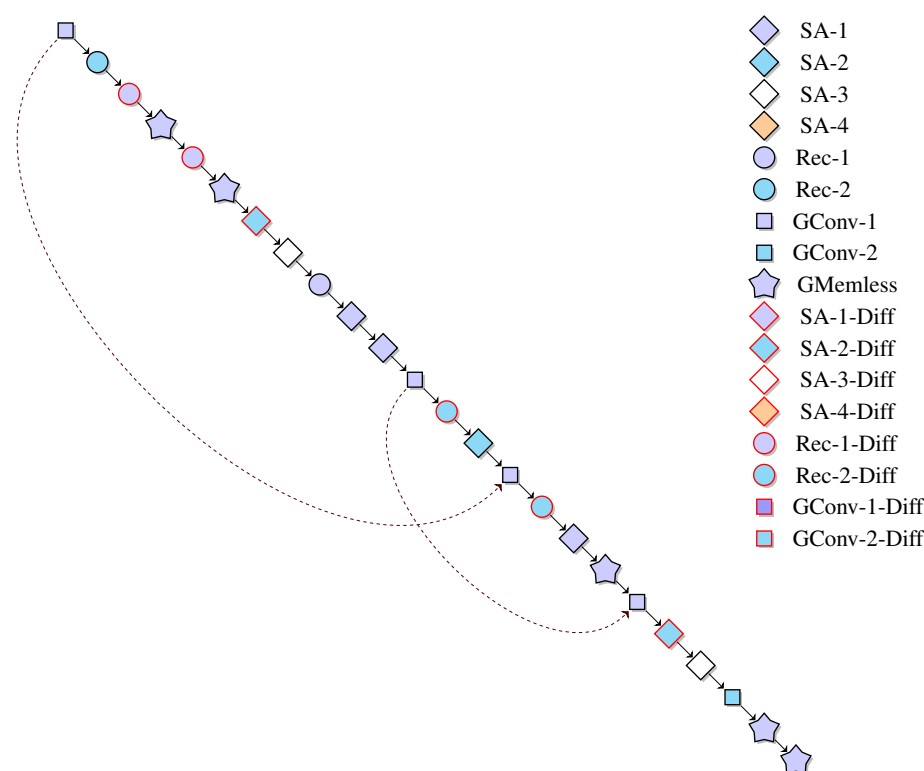

Figure B.12: STAR-4 optimised for quality and size (see Table 5.1). Dashed lines on the left indicate feature group sharing.

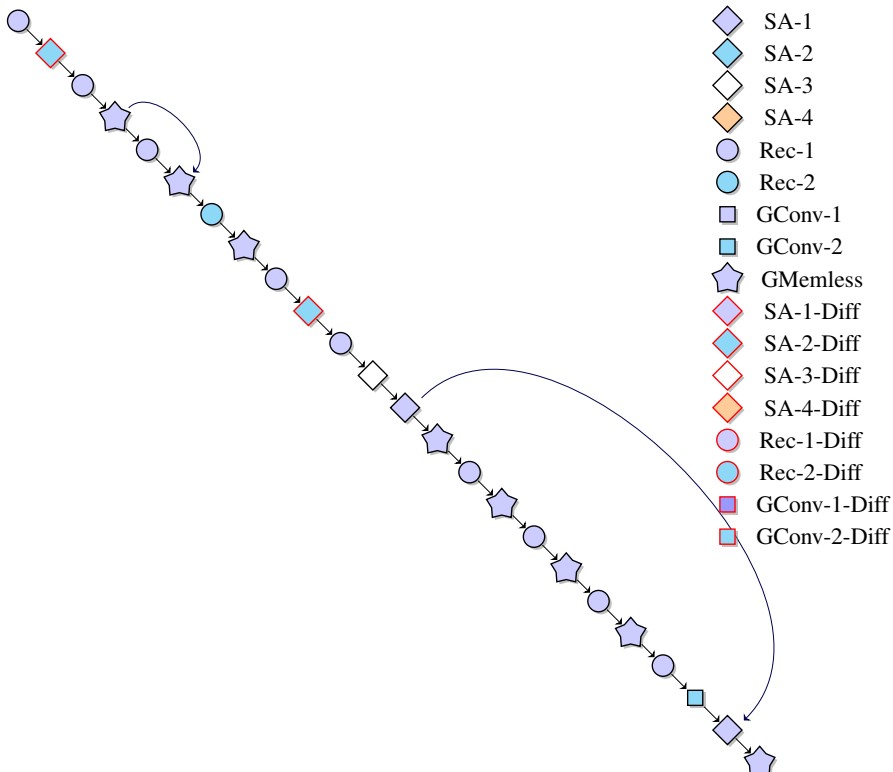

Figure B.13: STAR-5 optimised for quality and size (see Table A.4). Solid lines on the right indicate featurizer sharing.

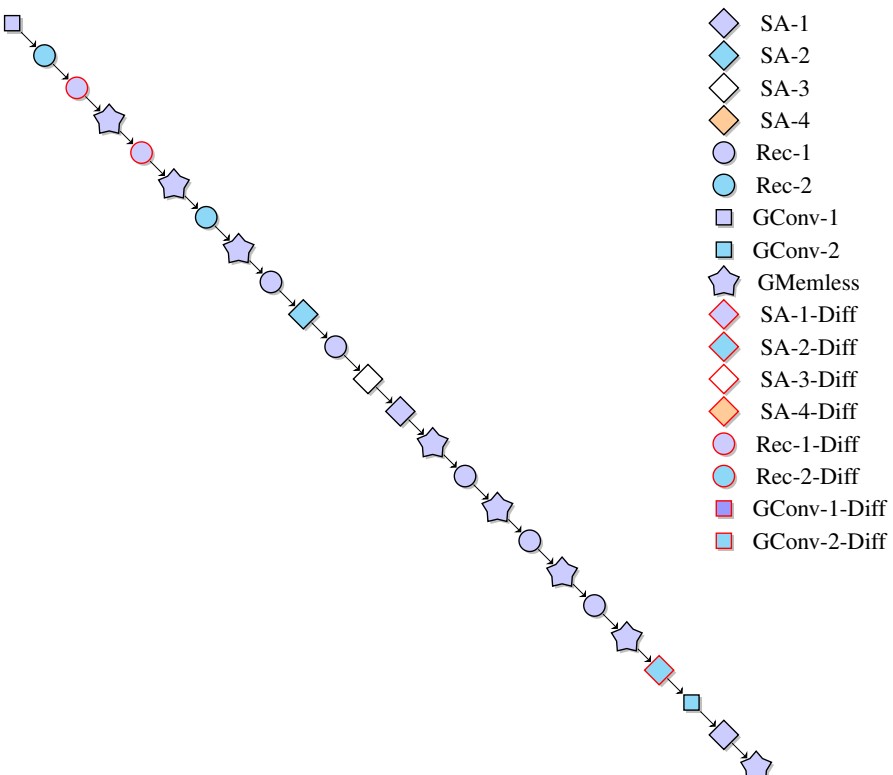

Figure B.14: STAR-6 optimised for quality and size (see Table A.4).

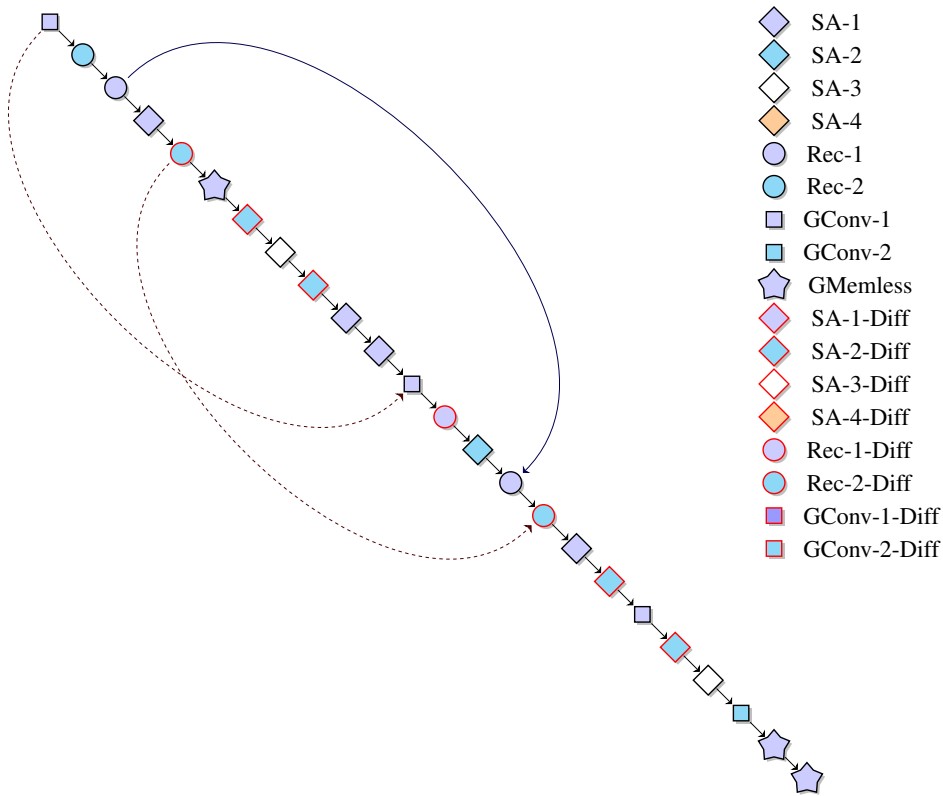

Figure B.15: STAR-7 optimised for quality and size (see Table A.4). Dashed lines on the left indicate feature group sharing while solid lines on the right indicate featurizer sharing.

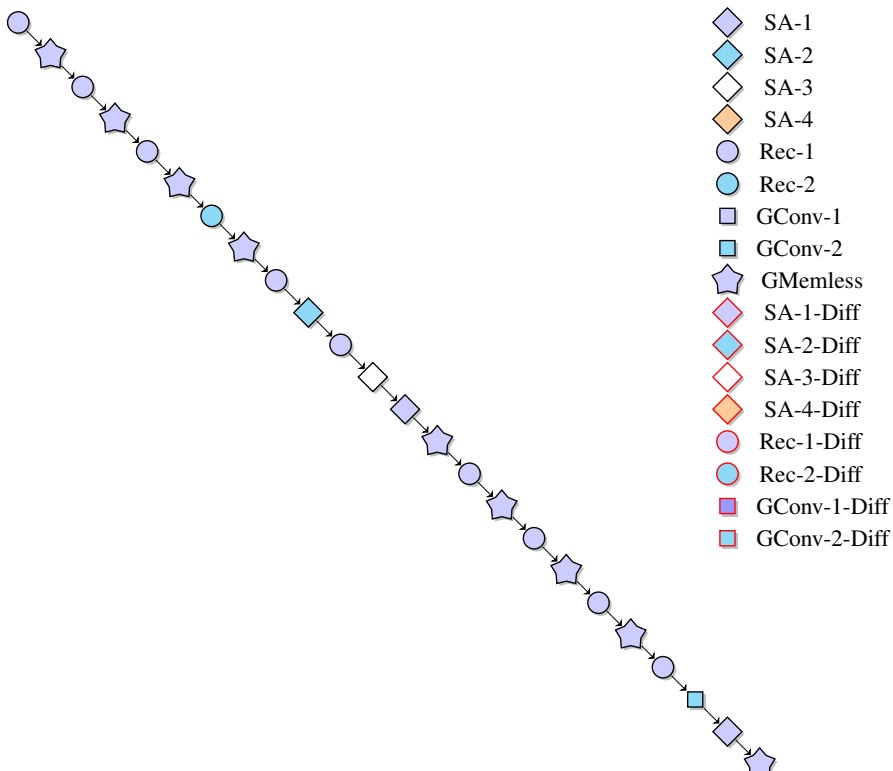

Figure B.16: STAR-8 optimised for quality and size (see Table A.4).

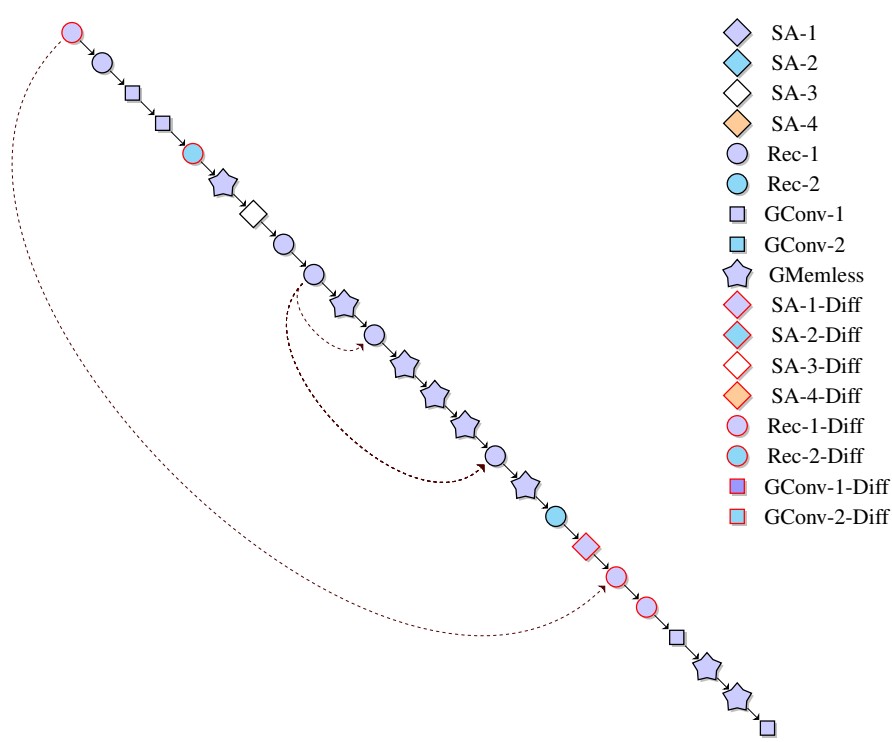

Figure B.17: STAR-1 optimised for quality and cache (see Table 5.1). Dashed lines on the left indicate feature group sharing.

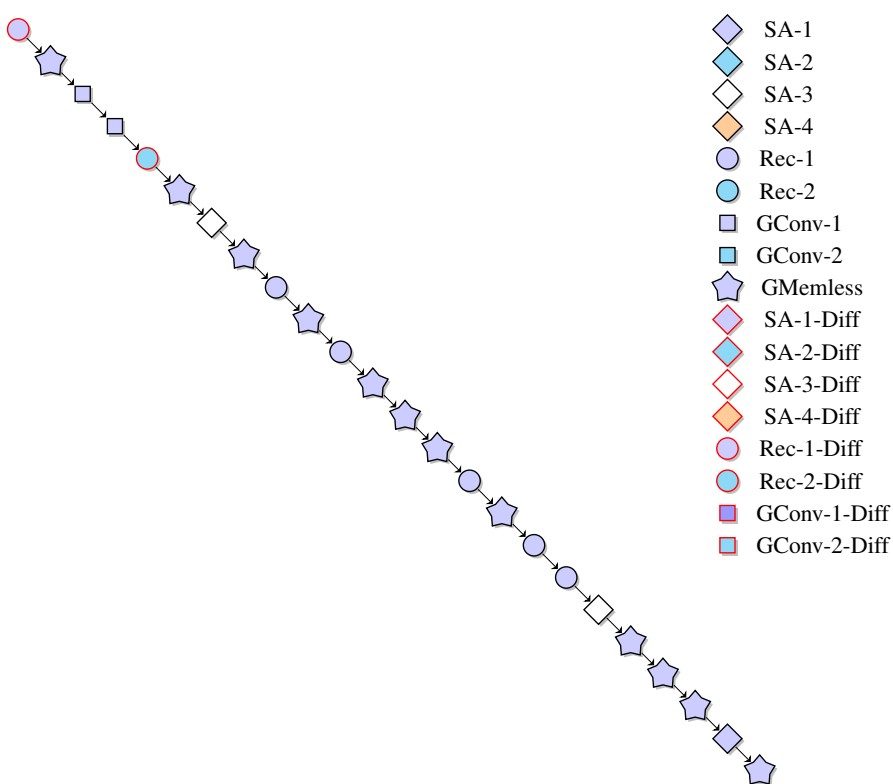

Figure B.18: STAR-2 optimised for quality and cache (see Table 5.1).

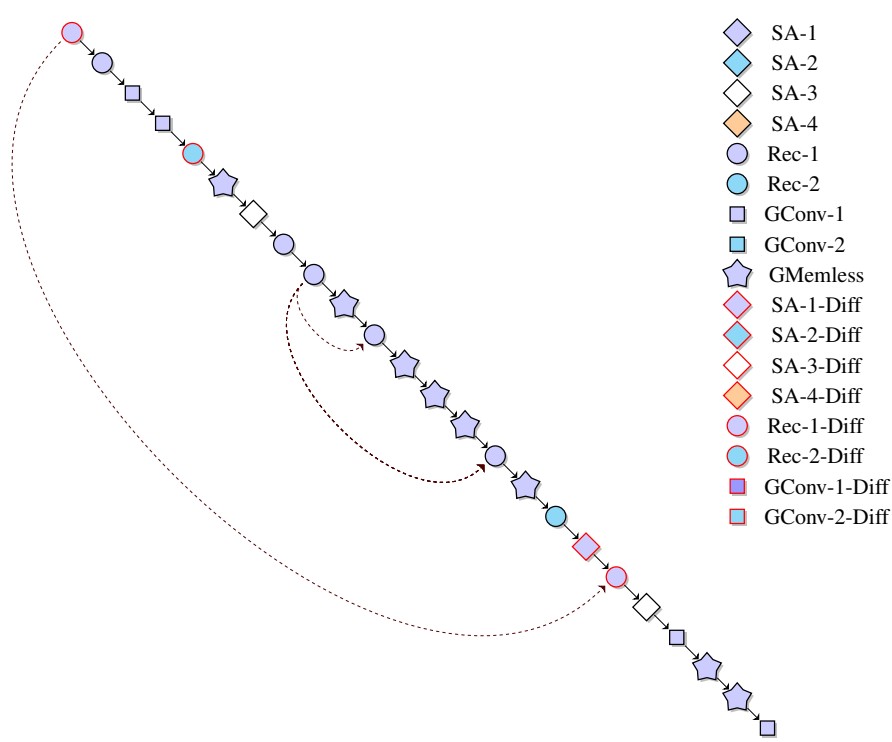

Figure B.19: STAR-3 optimised for quality and cache (see Table 5.1). Dashed lines on the left indicate feature group sharing.

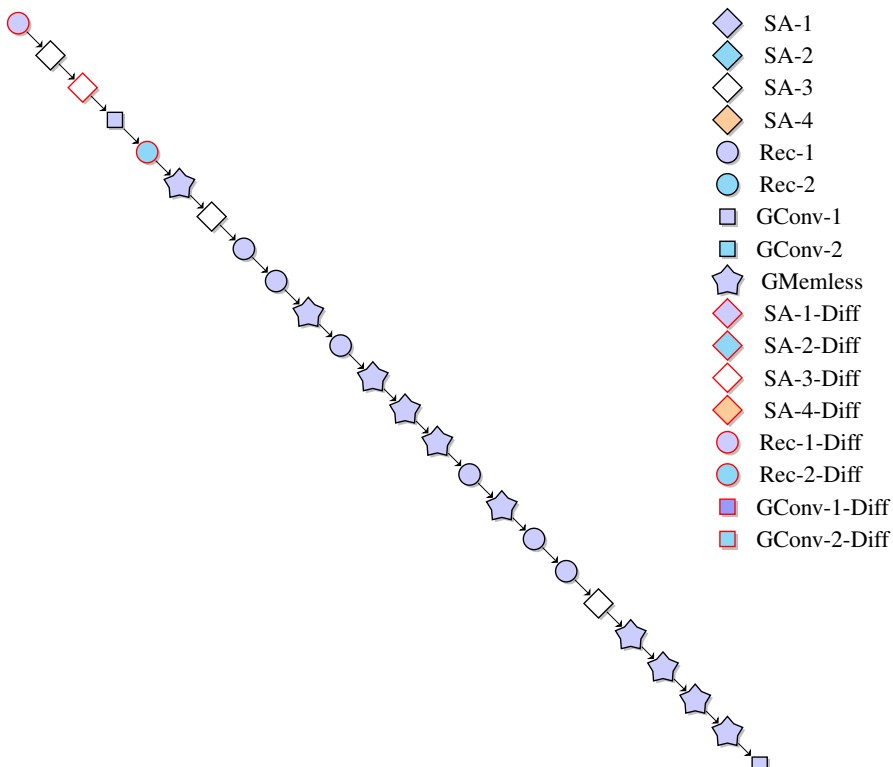

Figure B.20: STAR-4 optimised for quality and cache (see Table 5.1).

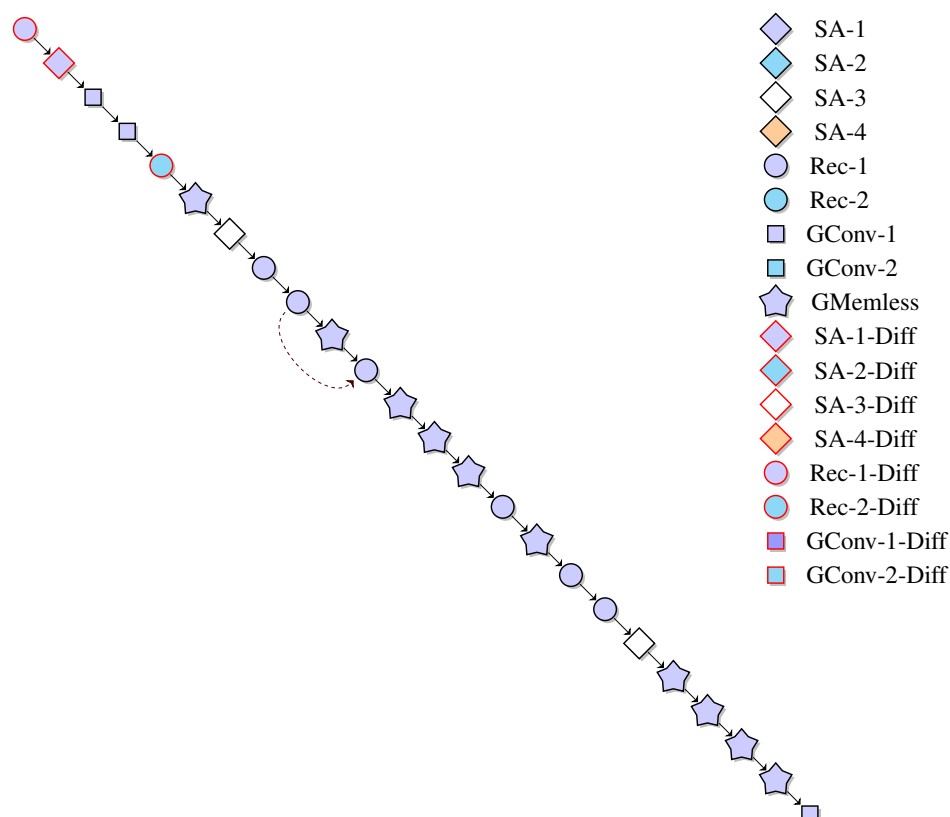

Figure B.21: STAR-5 optimised for quality and cache (see Table A.4). Dashed lines on the left indicate feature group sharing.

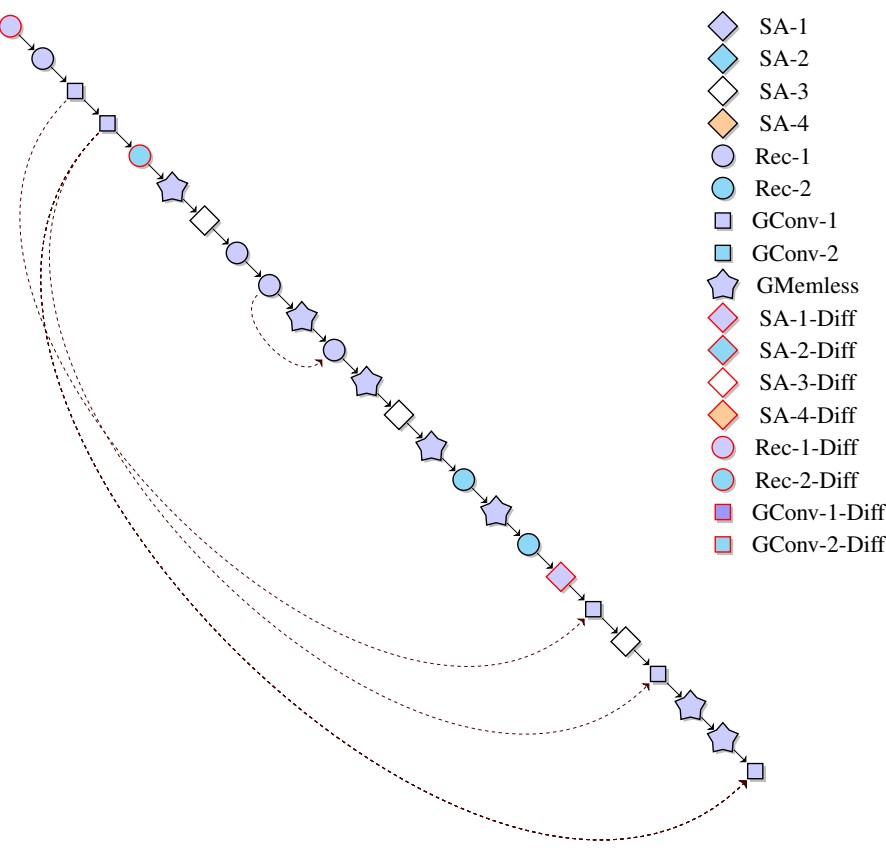

Figure B.22: STAR-6 optimised for quality and cache (see Table A.4). Dashed lines on the left indicate feature group sharing.

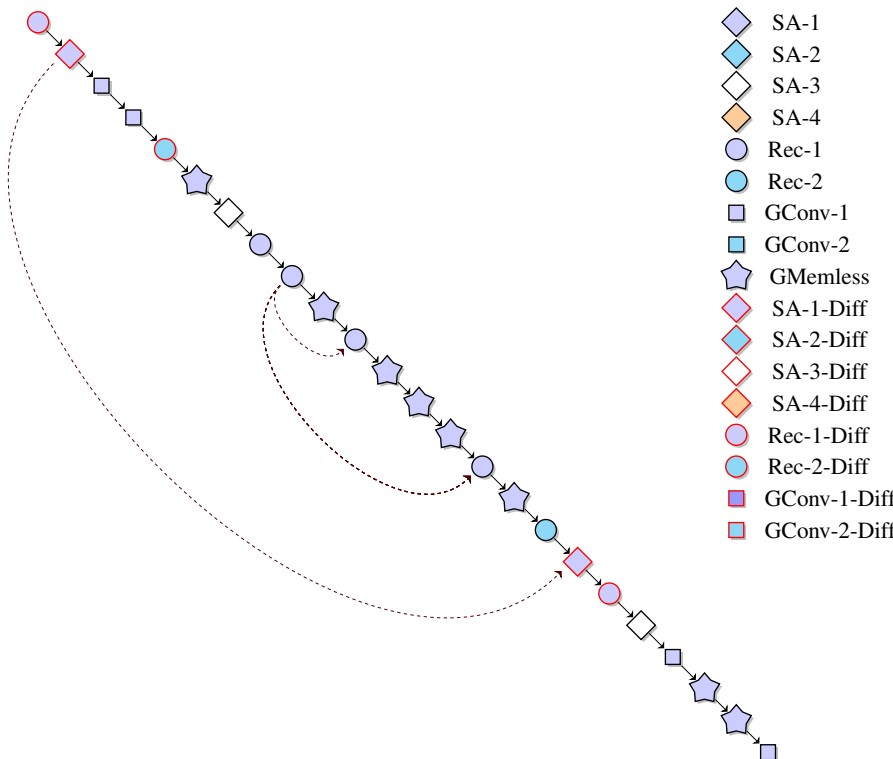

Figure B.23: `STAR`-7 optimised for quality and cache (see Table A.4). Dashed lines on the left indicate feature group sharing.

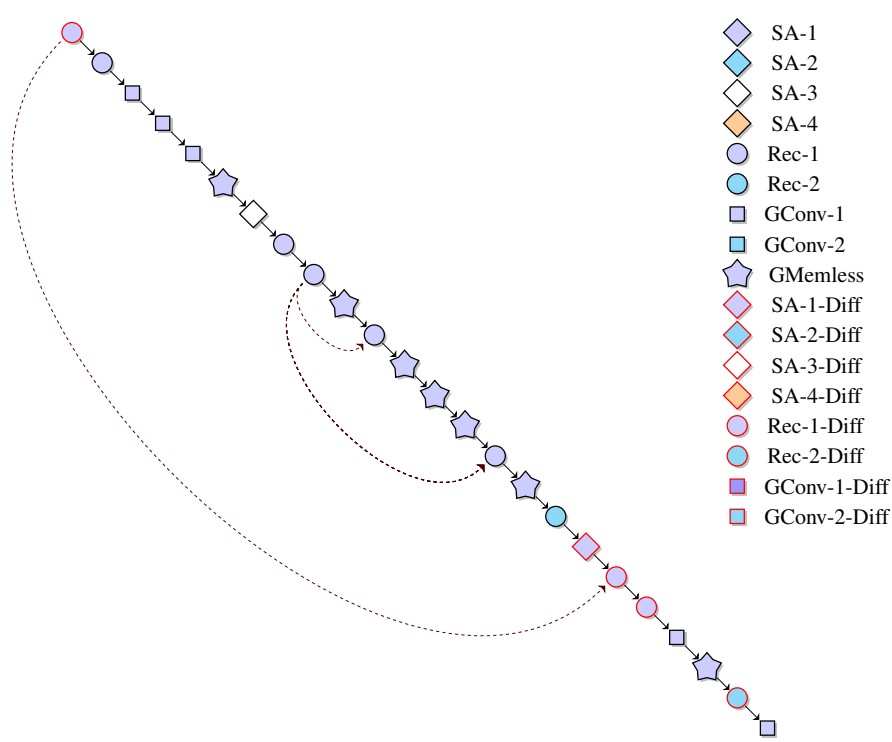

Figure B.24: STAR-8 optimised for quality and cache (see Table A.4). Dashed lines on the left indicate feature group sharing.

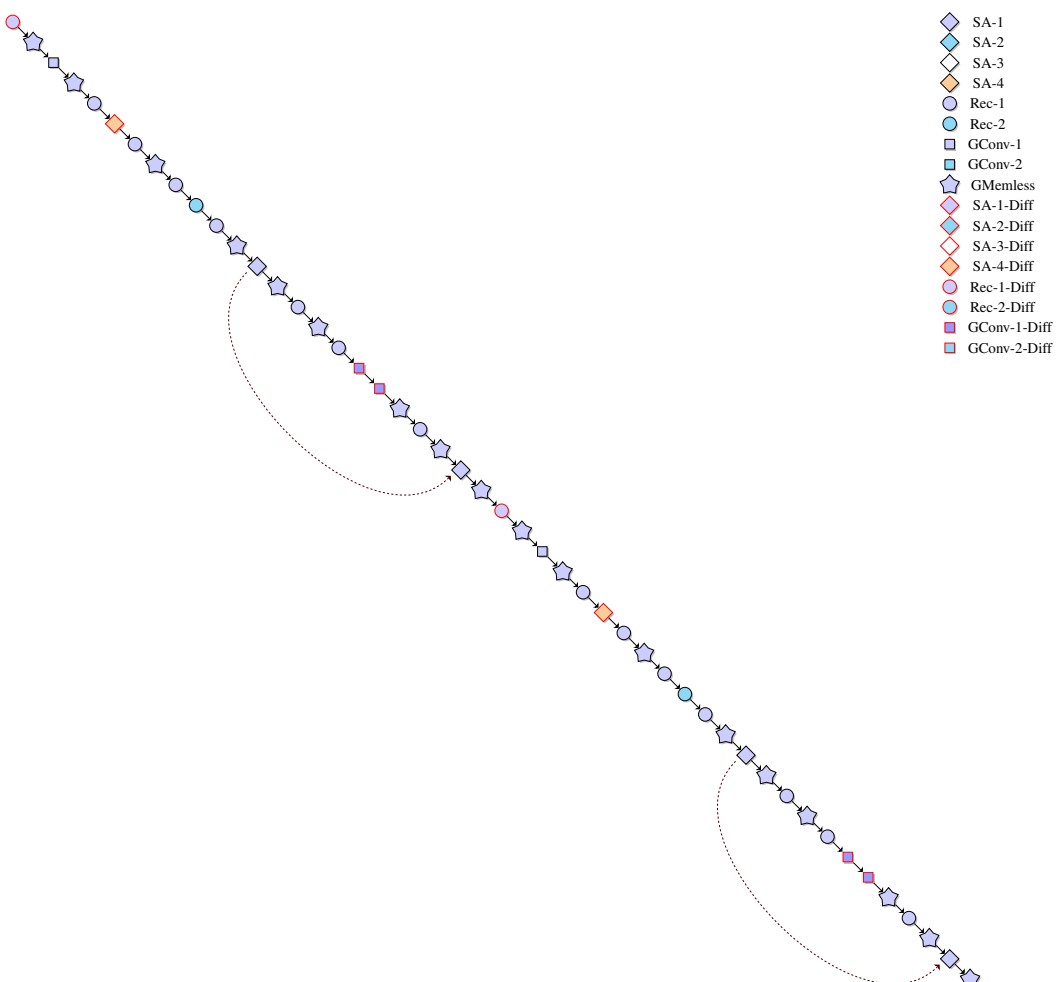

Figure B.25: STAR backbone optimized for quality and cache, consisting of 48 LIVs with a width of 2048 dimensions (see Table 5.2). This backbone was generated by duplicating a backbone from the STAR evolution for quality and cache and increasing its width from 768 to 2048 dimensions. Dashed lines on the left indicate feature group sharing.

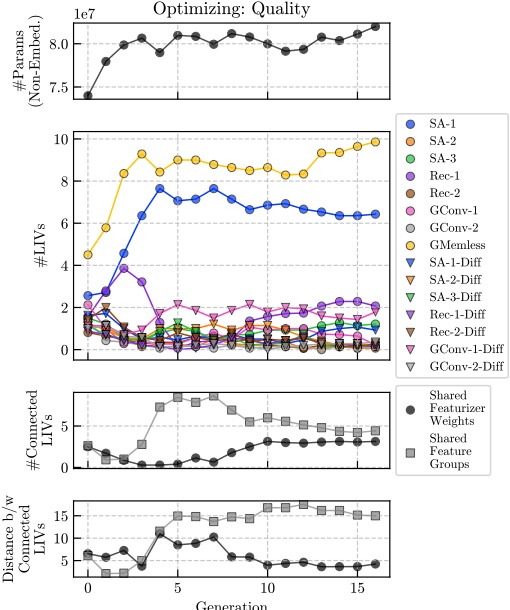

Figure B.26

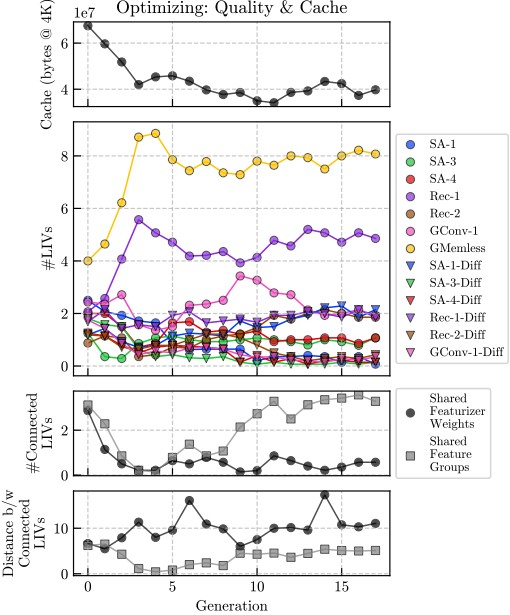

Figure B.27

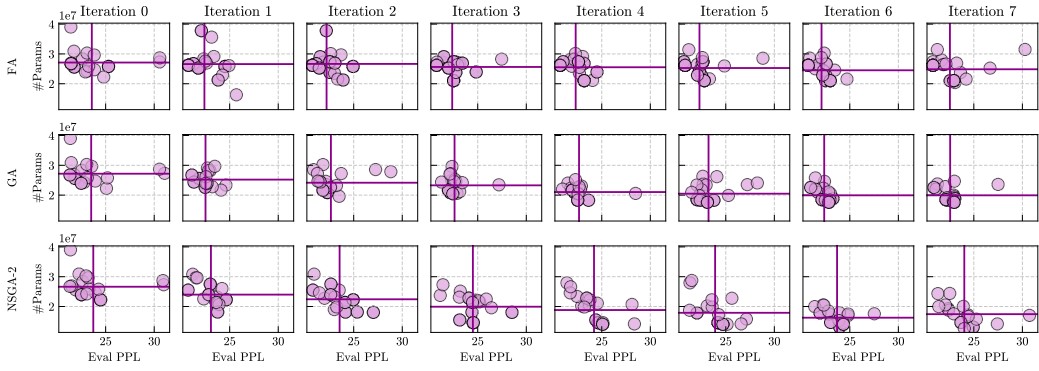

Figure B.28

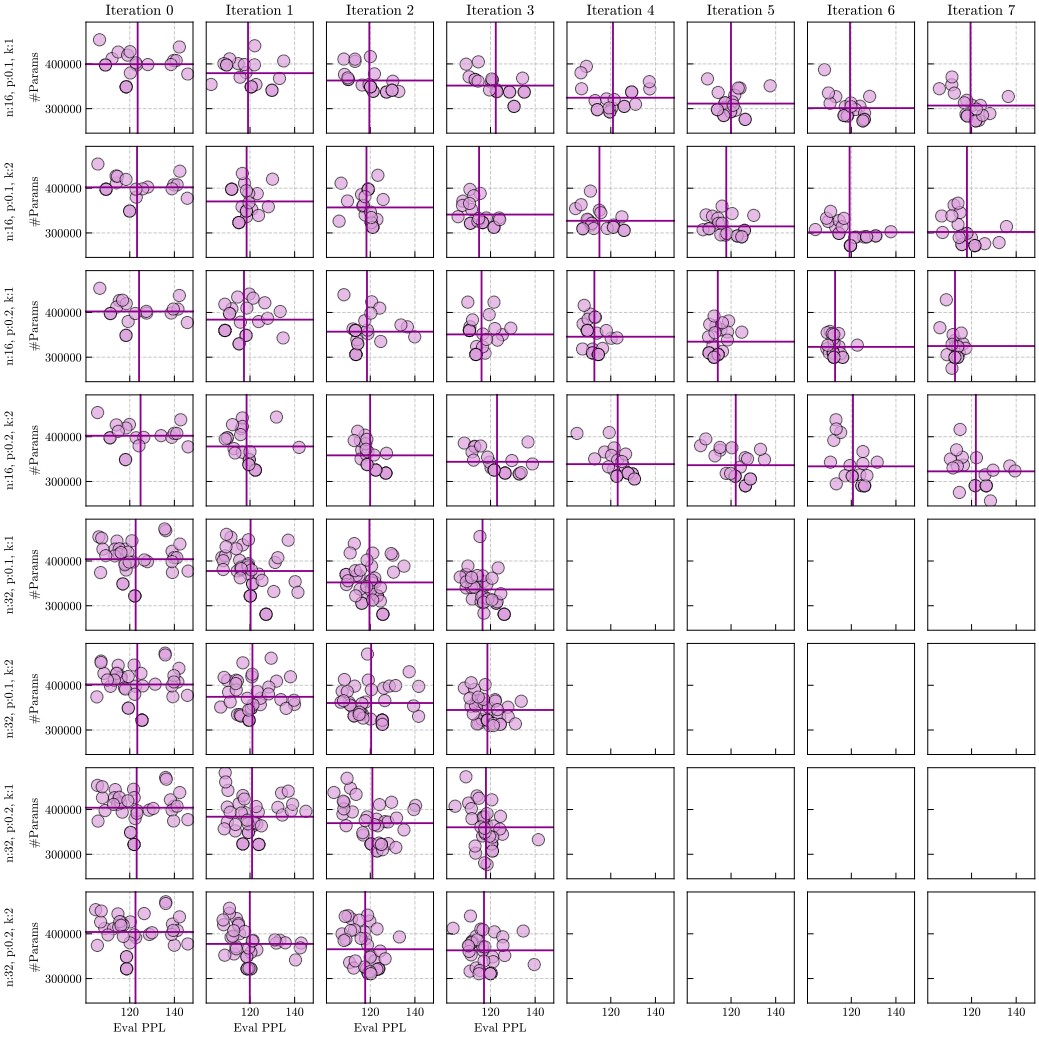

Figure B.29

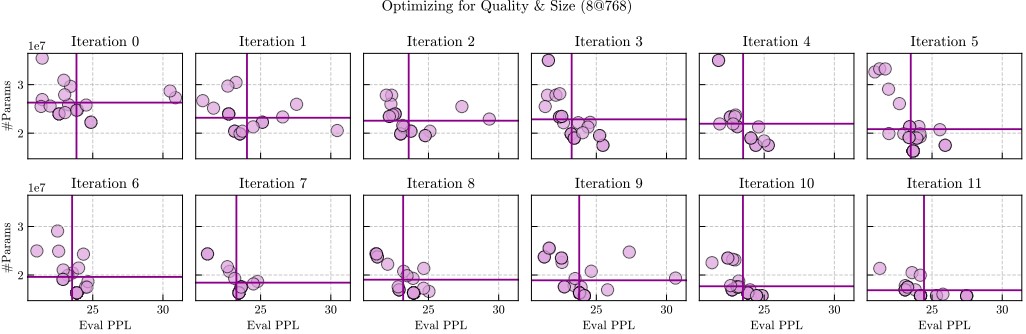

Figure B.30

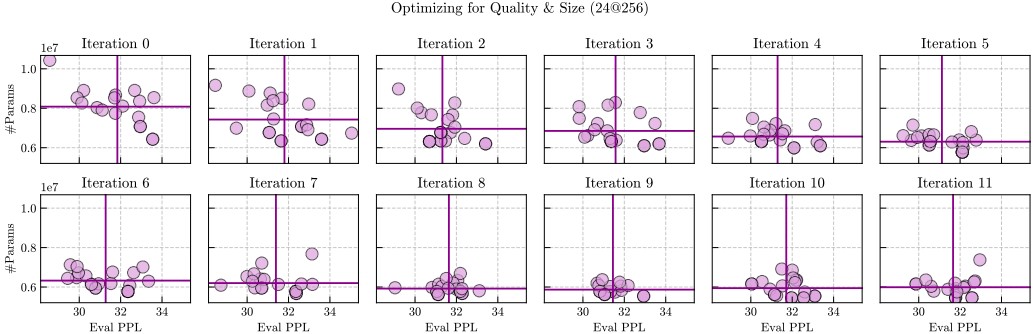

Figure B.31

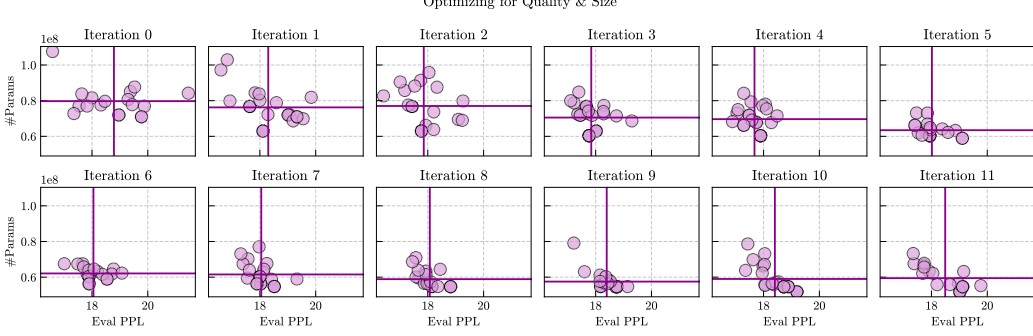

Figure B.32: Note that we used different evaluation datasets for our ablation and main experiments.

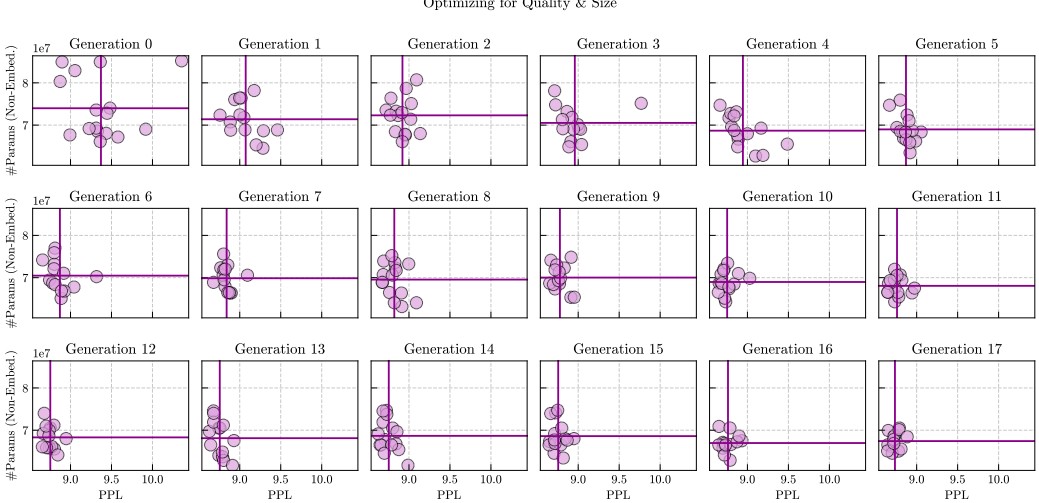

Figure B.33

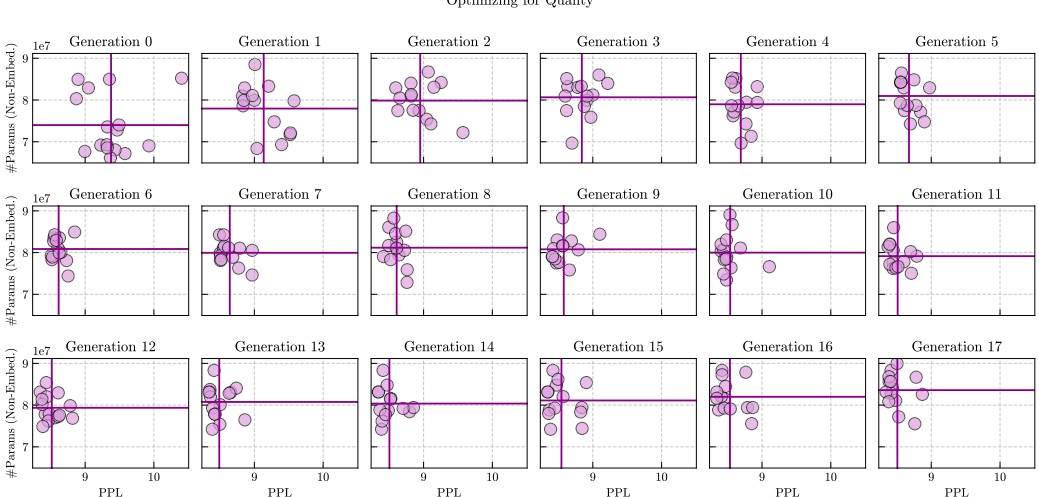

Figure B.34

Figure B.35

