# OpenReview forum: "STAR: Synthesis of Tailored Architectures"
_ICLR.cc/2025/Conference — ICLR 2025 Oral_

### Official Review · Reviewer_mX7E · 2024-11-02

**Soundness:** 2
**Presentation:** 3
**Contribution:** 2
**Rating:** 6
**Confidence:** 4

**Summary:**

This paper proposes the approach for the synthesis of tailored architectures (STAR). STAR genomes are automatically refined and recombined with gradient-free, evolutionary algorithms to optimize for multiple model quality and efficiency metrics. In the experimental results, the proposed method is compared with Transformer++ and StripedMamba.

**Strengths:**

1.	The proposed method combines a novel search space based on the theory of linear input-varying systems, supporting a hierarchical numerical encoding into architecture genomes.
2.	The proposed method STAR optimizes architectures for various combinations of metrics simultaneously: quality (perplexity during pretraining), quality and size, and quality and inference cache size (KV cache and fixed state cache).
3.	The experimental results are well analyzed.

**Weaknesses:**

The proposed method is only compared with Transformer++ and StripedMamba. Comparison with additional LLMs (such as Llama series) would be preferred.

**Questions:**

1.	How about the performance compared with other LLMs, such as Llama series?
2.	How about the performance compared with KV cache optimization methods ?

---

> ### Author Response · Authors · 2024-11-25
>
> Thank you for your thoughtful review and for highlighting the strengths of our work. We appreciate your recognition of the novel aspects of our approach, including the use of a hierarchical search space based on the theory of linear input-varying systems and the simultaneous optimization of multiple metrics. We're also glad that you found our experimental analysis to be well-analyzed. Below, we address your comments and provide further clarification:
>
> **Weakness 1 / Question 1: Comparison to pre-trained LLMs such as those of the Llama series.**
>
> Thank you for this insightful suggestion. STAR is designed to develop deep learning architectures tailored to specific objectives. To evaluate candidate architectures, and to provide a fair comparison, all architectures–baselines and STAR synthesized–are parameter-matched and trained from scratch on the same dataset. We emphasize that the baseline Transformer++ architecture in our paper is identical to the architecture used by the Llama model series. The StripedMamba baseline is based on Jamba (Lieber et al., 2024), another state-of-the-art pre-trained LLM.
>
> We have further extended our analyses to include comparisons at the 1B-parameter scale, showing that a STAR-synthesized backbone optimized for quality and cache size outperforms a parameter-matched Transformer++ baseline and matches a StripedMamba baseline in quality when trained for 40B tokens on the same data—while reducing cache size by 90% and 37%, respectively. For this analysis, we scaled a 125M-parameter STAR backbone by increasing depth from 24 to 48 operators and width from 768 to 2048 dimensions. This finding suggests that the advantages of STAR synthesized backbones observed at smaller scales translate to larger model scales akin to those of the referenced pre-trained LLMs.
>
> **Question 2: Comparison to other KV-cache optimization methods**
>
> Thank you for this valuable question. Our results demonstrate that STAR can design cache-optimized architectures that match Transformer++ and StripedMamba in quality while offering significant cache size reductions. This is achieved solely by optimizing the choice of operators and their compositions, such as feature group sharing (e.g., KV-cache sharing) and shared featurizer weights. Importantly, existing cache optimization methods like quantization can be applied to STAR-synthesized architectures just as they are to other architectures like Transformer++. These methods thereby provide additional tools to further reduce cache sizes for STAR models and are complementary rather than competitive approaches.
>
> **Reference:**
>
> Lieber, O., Lenz, B., Bata, H., Cohen, G., Osin, J., Dalmedigos, I., ... & Shoham, Y. (2024). Jamba: A hybrid transformer-mamba language model. arXiv preprint arXiv:2403.19887.

---

### Official Review · Reviewer_uJi5 · 2024-11-03

**Soundness:** 3
**Presentation:** 4
**Contribution:** 3
**Rating:** 8
**Confidence:** 4

**Summary:**

The paper tackles a crucial research problem in the ML community—Optimizing the design of neural network architectures beyond predefined and typical search spaces. The authors posit important research questions on the limitations of existing automated neural network design approaches—mainly relying on predefined backbones and limited search parameters. Also, existing works on NAS methods overlook the inherent hierarchy of the NN design at the level of operators and architecture topology. The authors propose STAR, a novel framework to address the issues above, and build a synthetic, multi-pattern, hierarchical search space-optimized with evolutionary algorithms. A comprehensive evaluation has demonstrated the effectiveness of STAR, showing significant improvements in perplexity, downstream benchmarks, model size, and inference caching compared to optimized hybrid Transformer baselines.

**Strengths:**

- The paper presents an important research problem for the ML design automation community: The limitations of existing search spaces, especially for hybrid and transformer models in language processing applications, which typically require significant computational resources to evaluate different design choices.
- The hierarchical search space design is grounded in the theory of linear input-varying systems, providing a solid theoretical foundation. Characterizing neural network architectures through featurization, structure, and backbone offers a clear and logical approach, supporting the creation of a more generalizable search space that extends beyond predefined backbones.
- The authors' choice of an evolutionary algorithm to explore the hierarchical design space is well-justified, given the complexity of the architectures being searched. This approach facilitates the evolution of large populations of architectures using mutation and recombination operators.
- The evaluation section is thorough and detailed, with promising results that demonstrate the effectiveness of the STAR framework. Overall, the paper is well-written and flows smoothly.

**Weaknesses:**

- While Section 2 provides a thorough overview of the hierarchical search space, certain design choices would benefit from additional explanation, particularly in relation to existing implementations and empirical justification for their inclusion. For example, the distinction between channel and token mixing could be clarified, along with an explanation of the LIVs to which each can be applied. Since STAR operates within a hybrid design space that may incorporate either convolution or attention mechanisms, it would be helpful to discuss how each design choice applies to different LIV operators and whether any constraints are involved.
- Section 3 explains the STAR genome and the role of each gene. However, it lacks a detailed description of the encoding strategy used in the evolutionary process, particularly regarding parameter ranges and the significance of each genome gene. For example, it would be helpful to clarify which integers correspond to each LIV class and the specific meaning behind each integer value. Additionally, further explanation is needed on how the evolutionary algorithm processes the genome encoding. To enhance clarity, the authors are encouraged to include a table (in addition to Figure 3.1) listing all search parameters (genes), their definitions, possible values, and how they translate into the neural network architecture.
- In the abstract and introduction, the authors claim that STAR’s optimization framework is a gradient-free evolutionary algorithm. However, this statement is somewhat misleading, as certain parameters of the explored architectures are indeed trained during the STAR evolution process. To clarify, the authors should provide a detailed schematic of the optimization process, including a clear figure, and add specifics regarding the training and evaluation times within the STAR evolution. This would allow readers to better understand how much STAR improves upon native GA frameworks. For a gradient-free optimization, it would be interesting to see how zero-shot approaches (e.g., ZiCo [1]) can be integrated into the STAR framework to accelerate the fitness evaluation.
- Although Section 4.1 introduces the evolutionary operators, such as mutation and recombination, these are not explained in sufficient detail. Specifically, the authors do not clarify which types of mutation (e.g., uniform, polynomial) and recombination (e.g., uniform, fixed-point) operators are employed, nor do they discuss how the STAR framework manages invalid designs generated after evolution. Additionally, it would be helpful to understand the likelihood and extent to which STAR produces invalid designs during evolution. To address this important aspect, the authors should consider including a figure or algorithm that illustrates the evolution process, detailing how mutation and recombination are applied and explaining how invalid neural network designs are managed—whether through replacement, alteration, correction, or removal.
- Given the hierarchical nature of the design space and STAR's apparent use of a single-genome representation, concerns arise regarding optimization convergence. Specifically, what are the pros and cons of a one-shot optimization approach versus a multi-stage optimization strategy, where multiple evolutionary algorithms are employed to optimize different levels of the search space? The authors are encouraged to discuss this comparison and to evaluate STAR's scalability on complex tasks, where achieving convergence toward optimal architectures may be challenging.
- The evaluation of STAR is somewhat limited, as it is only applied to two architectures (i.e., Striped Mamba and Transformer++). Since this paper aims to push neural network design beyond typical search spaces, it would be beneficial to evaluate STAR models on state-of-the-art (SOTA) architectures that have demonstrated strong performance on benchmarks such as RedPajama, LAMBADA, HellaSwag, ARC-e, and PiQA. A comparison with SOTA models like LLama [2], BitNet [3], and Falcon-Mamba [4] would provide valuable insights. To ensure a fair comparison, all models could be trained under the same conditions, with both performance and efficiency metrics used to assess the effectiveness of STAR’s design approach.
- Reproducing STAR without access to the original source code proved challenging. Despite my best efforts to replicate the work using the information provided in the appendix, I was unable to do so. To facilitate further investigation into the technical implementation details, it is recommended that the authors share the source code.

**References:**
- [1]: Zuo, Jingwei, et al. "Falcon Mamba: The First Competitive Attention-free 7B Language Model." arXiv preprint arXiv:2410.05355 (2024).
- [2]: Touvron, Hugo, et al. "Llama: Open and efficient foundation language models." arXiv preprint arXiv:2302.13971 (2023).
- [3]: Ma, Shuming, et al. "The era of 1-bit llms: All large language models are in 1.58 bits." arXiv preprint arXiv:2402.17764 (2024).
- [4]: Zuo, Jingwei, et al. "Falcon Mamba: The First Competitive Attention-free 7B Language Model." arXiv preprint arXiv:2410.05355 (2024).

**Questions:**

- Could you provide additional explanation for certain design choices within the hierarchical search space in Section 2, specifically regarding how they relate to existing implementations and the empirical rationale for their inclusion?
- Since STAR operates within a hybrid design space that may involve either convolution or attention operations, could you discuss how each design choice applies to different LIV operators, including any constraints involved?
- Could you elaborate on the encoding strategy used in the evolutionary process, particularly in terms of parameter ranges and the significance of each genome gene?
- You mention in the abstract and introduction that STAR’s optimization framework is gradient-free, yet some parameters of the explored architectures are trained during the evolutionary process. Could you clarify this point?
- Could you provide more detailed information on the mutation (e.g., uniform, polynomial) and recombination (e.g., uniform, fixed-point) operators used within STAR?
- How does the STAR framework handle invalid designs generated through evolution, and could you provide information on the likelihood and extent of these occurrences?
- Given that STAR operates within a hierarchical design space with a one-genome representation, there are some concerns about optimization convergence. Could you discuss the pros and cons of using a one-shot optimization approach versus a multi-stage optimization strategy that involves separate evolutionary algorithms for different levels of the search space?
- Without access to the original source code, reproducing STAR is challenging. Would you be open to sharing the source code to allow for further investigation into the technical details of your implementation?
- The current evaluation seems limited to only two architectures (Striped Mamba and Transformer++). Since this paper aims to explore neural network design beyond typical search spaces, would you consider evaluating STAR models on state-of-the-art architectures that perform well on benchmarks such as RedPajama, LAMBADA, HellaSwag, ARC-e, and PiQA?
- A comparison between STAR models and other SOTA models, such as LLama, BitNet, and Falcon-Mamba, trained under consistent configurations, could offer valuable insights. Would you consider including performance and efficiency metrics to assess STAR’s effectiveness relative to these models?

---

> ### Author Response · Authors · 2024-11-25
> **Response 1/2**
>
> Thank you for your thoughtful review and for highlighting the strengths of our work. We appreciate your recognition of the significance of the research problem we are tackling, that you view our hierarchical design space as clear and theoretically well-grounded, and find our choice of evolutionary optimization to search this space as well-justified. We're also glad that you found our experimental analysis thorough and the results promising. Below, we address your comments and provide further clarification:
>
> **Weakness 1 / Questions 1–2: Clarification on search space design choices and their relation to existing implementations.**
>
> Thank you for these valuable suggestions. Our revised manuscript clearly distinguishes between token- and channel-mixing structure (Section 2). Additionally, we now provide a full description of the STAR genome—and thus the hierarchical design space—in Appendix A.6, outlining the channel- and token-mixing structures of each LIV and how each LIV class relates to existing implementations. We focused on this set of operators because they encompass all empirically relevant operator types currently used in state-of-the-art language modeling. We have further extended the design space to include novel differential variants of all applicable LIV classes, similar to the "Differential Transformer" (Ye et al., 2024) (see Section 5) .
>
> **Weakness 2 / Question 3: Clarification of the meaning and parameter ranges of each STAR genome entry.**
>
> Thank you for this insightful comment. Our revised manuscript now includes a full description of each entry in the STAR genome, the possible values they can take in our analyses, and how these translate to specific architectures (Appendix A.6). We believe this greatly enhances the reproducibility and clarity of our work.
>
> **Weakness 3 / Question 4: Clarification on why the evolutionary algorithms are considered gradient-free.**
>
> We thank the reviewer for raising this insightful question. In STAR, models are optimized for quality by evaluating and comparing their performance after being trained on a predefined dataset shared across all candidate architectures. The evolutionary optimization algorithm, which is used to optimize the genomes representing the trained architectures, relies solely on the performance score of the trained model, along with other metrics such as parameter count and cache size. Importantly, it does not access the gradients used during model training. For this reason, we refer to the evolutionary algorithms employed in STAR as gradient-free—a term commonly used in the evolutionary optimization literature. To clarify this for readers, we have updated Figures 1.1 and 4.1 to include detailed schematics of the optimization process, clearly illustrating the roles of model training and evaluation within the evolutionary framework.
>
> **Weakness 4 / Questions 5–6: Details on recombination, mutation operations, and repair of invalid genome entries.**
>
> Thank you for pointing out the need for clarity in this area. We have revised the manuscript to specify that we employ uniform random sampling from the set of valid options for mutations. The valid options for each genome entry are detailed in Appendix A.6. We use the k-point crossover method for recombination, outlined in detail in Appendix A.4. As you correctly noted, these operations can result in invalid genome entries. Our revised manuscript now clearly explains how we repair any invalid entries that may occur in the backbone genome (Section 4.2). Specifically, if mutations or recombinations result in invalid configurations—such as incompatible sharing strategies—we detect and repair them by removing the invalid connection for entries 2 and 4 of the backbone genome or by re-sampling valid values for entries 1, 3, and 5. Importantly, these repair operations are infrequent during STAR optimization, occurring less than 20% of the time.

---

> ### Author Response · Authors · 2024-11-25
> **Response 2/2**
>
> **Weakness 5 / Question 7: Benefits and drawbacks of holistic versus multi-level genome optimization.**
>
> Thank you for this excellent question. We have expanded our discussion section (Appendix A.1) to address the advantages and drawbacks of treating the genome as a unified entity versus using a multi-stage optimization strategy targeting each hierarchical level. The holistic approach benefits rapid iteration and resource-limited scenarios but may not fully exploit the hierarchical design space, potentially overlooking dependencies across genome levels. In contrast, a multi-stage optimization strategy could improve convergence by leveraging the genome's modularity and addressing interactions incrementally, though it increases algorithmic complexity and computational costs. Notably, even with our current holistic approach, 90–100% of the synthesized architectures outperform baseline models on the respective metrics (quality, quality and size, quality and cache size), thus meeting our objectives.
>
> **Weakness 6 / Questions 9–10: Clarification on the choice of STAR baseline models.**
>
> Thank you for this insightful suggestion. STAR is designed to develop deep learning architectures tailored to specific objectives. To evaluate candidate architectures, we train them from scratch and compare them to parameter-matched, state-of-the-art baseline architectures trained on the same data. We emphasize that the baseline Transformer++ and StripedMamba architectures in our paper are identical to/based on those used in state-of-the-art pre-trained models like Llama, which uses a Transformer++ architecture, and Jamba (Lieber et al., 2024), which is based on a StripedMamba architecture. We did not include a pure Mamba baseline architecture, as employed by Falcon-Mamba, as it has been empirically shown to be inferior in quality to a respective striped variant incorporating a small set of attention operators (e.g., Poli et al., 2024). As suggested by the reviewer, we train parameter-matched variants of these architectures on the exact same data as STAR-synthesized backbones to ensure a fair comparison.
>
> We have further extended our analyses to include comparisons at the 1B-parameter scale, showing that a STAR-synthesized backbone optimized for quality and cache size outperforms a parameter-matched Transformer++ baseline and matches a StripedMamba baseline in quality when trained for 40B tokens on the same data—while reducing cache size by 90% and 37%, respectively. For this analysis, we scaled a 125M-parameter STAR backbone by increasing depth from 24 to 48 operators and width from 768 to 2048 dimensions. This demonstrates that the advantages observed at smaller scales translate to larger models akin to those referenced.
>
> **Weakness 7 / Question 8: Improving the reproducibility of STAR.**
>
> Thank you for highlighting this important aspect. Our revised manuscript now provides a full description of all entries in the STAR genome, the possible values they can take, and how these relate to existing algorithm implementations (Appendix A.6). We also clearly outline each step of the evolutionary optimization process, detailing how genomes are mutated, recombined, and how we repair any invalid entries (Section 4 and Appendix A.4). We believe these updates significantly enhance the reproducibility of our work.
>
>
> **References:**
>
> - Ye, T., Dong, L., Xia, Y., Sun, Y., Zhu, Y., Huang, G., & Wei, F. (2024). Differential transformer. arXiv preprint arXiv:2410.05258.
> - Lieber, O., Lenz, B., Bata, H., Cohen, G., Osin, J., Dalmedigos, I., ... & Shoham, Y. (2024). Jamba: A hybrid transformer-mamba language model. arXiv preprint arXiv:2403.19887.
> - Poli, M., Thomas, A. W., Nguyen, E., Ponnusamy, P., Deiseroth, B., Kersting, K., ... & Massaroli, S. (2024). Mechanistic design and scaling of hybrid architectures. arXiv preprint arXiv:2403.17844

---

> ### Comment · Reviewer_uJi5 · 2024-11-26
>
> I appreciate the authors' comprehensive and well-articulated responses, which have successfully addressed several of my concerns. I strongly encourage the authors to integrate these clarifications into the revised manuscript to further strengthen its presentation. Sharing the codebase would also be a valuable addition, promoting transparency and enabling reproducibility. I strongly recommend the authors to open-source their code in the revised version of the paper. Also, there are missing graphs in Figure B.29, the authors should update this figure by including all of the graphs. Overall, Considering the significance of the paper's contributions and findings, specifically in neural networks design automation I am happy to revise my score and recommend the paper for acceptance.

---

### Official Review · Reviewer_hz6S · 2024-11-03

**Soundness:** 3
**Presentation:** 4
**Contribution:** 3
**Rating:** 8
**Confidence:** 3

**Summary:**

The authors propose a new approach for automated deep learning architecture search based on evolutionary optimization. To realize effective optimization, the work introduces 'linear input-varying systems' (LIVs), a general formulation of input-dependent token vector transformations. With this formalism, a wide range of commonly used model operators such as attention, recurrence, and convolutions can be expressed as special LIV-cases, that compose into model architecture backbones via a set of compositional rules. Notably, the set of LIVs and their composition can be expressed in a hierarchical coding scheme that describes a specific network architecture as a sequence of integers. This 'genome' lends itself to evolutionary optimization via recombination and random mutation of the genome integer elements. Using the LIV genome representation with multi-objective evolutionary algorithms such as NSGA-2, the authors demonstrate that they can automatically discover network architectures that offer improvements over highly optimized Transformer architecture baselines.

**Strengths:**

The work proposes a straightforward method for constructing a well-conditioned and comprehensive model architecture search space that is amenable to evolutionary optimization under multiple objectives. The hierarchical construction of the LIV building blocks and the coding schemes ensures that a wide variety of candidate architectures can be expressed and searched efficiently. Notably, the genome representation has a clear interpretation which makes it easy to apply constraints to ensure robustness to random edits.

The paper is well-written and provides a clear presentation of the motivation and context of this work. The experimental evaluation is logically structured and presents the key findings in an accessible way.

The extensive experimental results provide convincing evidence that the STAR method is effective in finding high-performing architectures under various optimization objectives.

**Weaknesses:**

By construction, the method appears to be geared towards finding tweaks of transformer-based architecture stacks and does not aim to discover fundamentally new architecture designs. For instance, genomes are constructed assuming a pre-norm residual structure that cannot be varied by the optimizer. This is a reasonable assumption but it arguably relies on the manual design expertise that automated architecture search is ultimately aiming to overcome. Can you think of ways to extend the genome to parametrize the residual connection and normalization schemes?

On a similar note, the mutation constraints described in Section 4.2 -- while well motivated and reasonable -- suggest that an unconstrained STAR genome is not sufficiently robust to random edits, meaning that expert intuition is still required to 'guide' the mutation process. Can you quantify the impact of unconstrained edits on the search outcomes, e.g. through an ablation study that compares constraint vs unconstrained mutations?

Finally, minor points, I found the notation in Eq. L128 and L138 a little confusing.
- The underbrace in Eq. L128 suggests that the summation is part of $T_{ij}^{\alpha \beta}$, but in Eq. L138 the summation is repeated.
- Additionally, as noted in L155 $T_{ij}$ is used to denote a whole range of possible operators, so it seems that $T_{ij}^{\alpha \beta}$ in L128 is overloaded. It may be clearer to use the underbrace in Eq. L128 to annotate 'attention $T_{ij}$' and 'attention $T^{\alpha \beta}$' explicitly as the special case and then use $T_{ij}^{\alpha \beta}$ to define the generalized version in Eq. L138.

**Questions:**

I might have missed it, but in Table 5.1, what is the difference between STAR-1 to STAR-8? Presumably, the optimization finds a Pareto-front of solutions, but I couldn't find any details on how you picked the evaluated solutions. Could you please provide details on the selection criteria for these eight models and explain how they differ architecturally? This would give readers a clearer picture of the diversity of solutions found by the method.

The experiments used a fixed number of LIVs to form a backbone. How did you choose this default backbone length of 24 LIVs? Could you discuss the potential challenges and benefits of allowing a variable-length search space with an arbitrary number of LIV modules, and whether this would be something to consider for future work?

---

> ### Author Response · Authors · 2024-11-25
>
> Thank you for your thoughtful review and for highlighting the strengths of our work. We appreciate your recognition of the empirical strengths, versatility, and efficiency of our hierarchical design space, and that you find our paper well-written and clear. We're also glad that you found our experimental analysis to provide convincing evidence in favor of the STAR’s ability to effectively synthesize high-quality architectures tailored to various objectives. Below, we address your comments and provide further clarification:
>
> **Weakness 1: Synthesizing improved residual streams.**
>
> Thank you for this valuable suggestion. We have extended the backbone genome to support residual streams beyond the conventional pre-norm structure. Appendix A.7 details this extension and includes an ablation study comparing STAR with and without it. The results show that this extension enables STAR to synthesize architectures with slightly smaller parameter counts while maintaining the same quality. Based on these promising findings, we plan to further explore residual composition strategies in future work.
>
> **Weakness 2: Does the mutation process require expert intuition?**
>
> Thank you for this insightful comment. We have not observed any synthesized architectures exhibiting unstable training behavior. While STAR corrects invalid genome edits during evolution, these corrections are fully determined by the STAR genome itself and do not require expert knowledge, since every position of the genome has a specific function that does not change no matter to which objectives STAR is applied. To clarify this, we have added a complete description of the STAR genome and all its possible values in Appendix A.6. We have also enhanced our explanation of how invalid genome edits are repaired in the main text (Section 4.2).
>
> **Weakness 3: Underbrace in the attention operator math.**
>
> Thank you for the helpful suggestion. We have incorporated all of the reviewer's recommendations into our notation and have updated the math notation throughout to improve overall clarity for the reader.
>
> **Question 1: What are the differences between STAR 1–8, and how were they selected?**
>
> Thank you for these valuable questions. We have now included visualizations of all evaluated STAR backbones—eight per STAR evolution—in Figures B.1-25 and expanded our discussion of recurring architecture motifs (Appendix B.1). We selected these backbones from the pool of synthesized candidates as follows: after the evolution process, we chose the eight backbones that achieved the lowest perplexity among those with lower parameter counts (for the quality and quality-size optimizations) or smaller cache sizes (for the quality-cache optimization) compared to our baseline models. We now explicitly state this selection criterion in the main text (Section 5) to clarify it for readers.
>
> **Question 2: Why do you synthesize at a fixed depth of 24 LIVs?**
>
> We selected a depth of 24 LIVs at a width of 768 dimensions to match the commonly used "125M-parameter" model scale in language modeling experiments. Typically, each of the 12 layers in a 125M architecture comprises a sequence-mixing operator (e.g., attention) and a channel-mixing operator (e.g., SwiGLU), totaling 24 operators. While STAR can optimize variable-depth and variable-width architectures, this introduces challenges: shallower genomes are computationally cheaper and converge faster but may lack necessary complexity, whereas deeper genomes offer greater representational power but expand the search space, risking suboptimal convergence due to overfitting or inefficient sampling. In line with the reviewer's suggestion, we have added a new section discussing this topic (Appendix A.1).

---

> > ### Comment · Reviewer_hz6S · 2024-11-26
> >
> > Thank you for your detailed responses, they have answered my remaining questions. I appreciate the time the authors took to revise the manuscript with additional discussion and results; I find the manuscript much improved. My rating still stands, good paper!

---

### Official Review · Reviewer_GcDr · 2024-11-04

**Soundness:** 3
**Presentation:** 3
**Contribution:** 2
**Rating:** 6
**Confidence:** 3

**Summary:**

This work presents an iterative improvement of model architectures to achieve quality-efficiency frontier called STAR. It leverages the theory of  linear input-varying systems to obtain architecture genome encoding and optimize the model through evolutionary algorithms. The experiments reveal optimized models compared to highly optimized transformers and striped hybrid models.

**Strengths:**

--It shows promising results.
-The encoding for architecture genome seems to be novel.

**Weaknesses:**

-The idea of using evolutionary algorithm for the architecture design space exploration  is not new.

-The mutation mainly happens on the backbone and  it is not clear if will affect the underlying components.

**Questions:**

See the weaknesses.

---

> ### Author Response · Authors · 2024-11-25
>
> Thank you for your valuable review and feedback. We are pleased that you found the presented results promising and that you appreciated the novelty of our architecture genome encoding. Below, we address your comments and provide further clarification:
>
> **Weakness 1: Evolutionary algorithms for architecture design are not new.**
>
> We thank the reviewer for this insightful comment. We acknowledge that evolutionary algorithms (EAs) are well-established in architecture optimization. However, STAR distinguishes itself through a novel hierarchical search space for computational units and their composition, along with a unique numerical encoding–the STAR genome–tailored for evolutionary methods. The STAR design space, grounded in the theory of linear input-varying systems (LIVs), provides a new framework for creating architecture components. LIVs generalize key computational units in deep learning, including attention, recurrences, convolutions, and other structured operators. The framework characterizes architectures at three hierarchical levels: (a) featurization, modulating linear computation based on input context; (b) operator structure, describing the structure for token and channel mixing; and (c) backbone, detailing the interplay of LIVs. Unlike prior search spaces, the LIV search space is both comprehensive and stable, with most sampled candidates training successfully.
>
> **Weakness 2: Unclear which levels are affected by mutation.**
>
> We thank the reviewer for pointing out that it is unclear whether the mutation operations performed by STAR in fact affect the underlying LIV components. We would like to clarify that these mutations affect the specific characteristics of the LIVs and the way in which individual LIVs contained in a backbone are connected, for example, determining if they share specific feature groups (akin to KV-cache sharing), or weights. The mutation operation also alters the specific characteristics of the LIV classes, such as switching between softmax attention variants with varying KV-cache compression ratios (similar to multi-query attention). By incorporating these mutation strategies and focusing on optimizing the backbone genome, STAR is able to synthesize architectures that outperform strong Transformer++ and StripedMamba baselines on multiple objectives and downstream language modeling benchmarks.

---

### Author Response · Authors · 2024-11-25
**Common response to all reviewers**

We thank the reviewers for their constructive feedback, which greatly helped improve the clarity and quality of our paper. Below, we summarize key improvements of our revised manuscript:

- **Enhanced results throughout:** We updated all analyses with extended STAR evolutions, an improved LIV option pool, and differential variants of attention, gated convolutions, and recurrences (akin to the "Differential Transformer"). Now, 90-100% of evaluated synthesized architectures outperform parameter-matched Transformer++ and striped hybrid baselines (Tables 5.1 and A.4)
- **Scaling to 1B parameters:** We demonstrate that a 125M-parameter STAR backbone optimized for quality and cache size can scale to 1B parameters, matching or exceeding Transformer++ and hybrid baselines in quality while reducing cache size by 90% and 37%, respectively (Table 5.2).
- **Full genome details:** For reproducibility, we provide a full genome description, including all digits, their possible values, and meanings (Appendix A.6).
- **Updated methods figures:** To make our methodology more clear to the reader, we extended and updated Figures 1.1 and 4.1, which now clearly outline the various steps of STAR.
- **Visuals of all evaluated STAR backbones:** To give readers greater insight into the architectures resulting from STAR, we include a depiction of all evaluated STAR backbones (see Figures B.1-25) and extended our discussion of recurring architecture motifs (see Appendix B.1).
- **Synthesizing improved residual streams:** We tested a variant of STAR that also allows modeling new residual connections as part of the backbone genome, going beyond the previous pre-norm residual structure (Appendix A.7).
- **Extended discussion:** We now discuss the benefits and challenges of synthesizing variable-depth/width backbones and the trade-offs of holistic genome optimization versus targeted optimizations at multiple levels (Appendix A.1).

---

### Author Response · Authors · 2024-12-03
**Follow-Up and Clarifications Before Discussion Closure**

We thank the reviewers who have contributed to this discussion so far. As the discussion period draws to a close, we would like to revisit our earlier responses to ensure we have thoroughly addressed the questions posed by the remaining reviewers. If there are any additional questions or points requiring clarification, we are more than happy to provide further details.

---

### Meta-Review · Area_Chair_jXC4 · 2024-12-23

**Metareview:**

The paper introduces STAR (Synthesis of Tailored Architectures), a novel framework for neural network architecture search that uses hierarchical design spaces based on linear input-varying systems. It encodes architectures as genomes optimized via evolutionary algorithms to achieve superior trade-offs across multiple objectives, including model quality, size, and cache efficiency. STAR demonstrates strong empirical results, outperforming parameter-matched baselines like Transformer++ and StripedMamba in key metrics.
Reviewers praised the work for its novel search space design, solid experimental analysis, and clear presentation. Criticisms included limited baseline comparisons, the constrained genome structure, and insufficient explanation of the evolutionary operators. In response, the authors expanded the genome to support flexible residual streams, clarified mutation mechanisms, and extended evaluations to larger-scale architectures. I follow the reviewers' unanimous vote for acceptance. Since the work may be quite influential, I recommend it for a spotlight.

**Additional Comments On Reviewer Discussion:**

Two reviewers (GcDr and mX7E) never engaged after submitting their initial review (both giving scores 6). I weighted those reviews less. Reviewer uJi5 raised multiple issues, upon the clarification of which, they increased their score from 6 to 8.

---

### Decision · Program_Chairs · 2025-01-22

Accept (Oral)